# Copine-6 is a TRPM3 escort protein controlling the sensitivity of sensory neurons to noxious heat

Yiting Gao[1,4], Shengxiang Yan [1,4], Zhongyang Zhang[1], Jieyao Zhang[1], Meng Yang[1], Shihab Shah[2], Sofia Figoli[2], Qi Jing [3], Haixia Gao [1✉] & Nikita Gamper [1,2✉]

## Abstract

Copine-6 is a calcium-sensitive phospholipid-binding protein of the evolutionarily conserved Copine family. In the central nervous system, Copine-6 regulates function of some neurotransmitter receptors and structural plasticity of dendritic spines, influencing learning and memory. Copine-6 is expressed in peripheral somatosensory neurons, but its role in somatosensation remains unclear. Here we demonstrate that Copine-6 plays a prominent role in thermosensation. Copine-6 is highly expressed in a subpopulation of dorsal root ganglia (DRG) neurons that also express thermosensitive TRPM3 channels. Genetic deletion or downregulation of Copine-6 in the DRG in vivo selectively impairs sensitivity to noxious heat, without affecting other sensory modalities, and significantly reduced TRPM3 currents in DRG neurons. Copine-6 interacts with TRPM3 via its von Willebrand factor A (vWA) domain, promoting TRPM3 translocation to the plasma membrane. Thus, Copine-6-dependent TRPM3 trafficking determines noxious-heat sensitivity range of the nerve fibers; moreover, Copine-6 is an accessible target for the treatment of heat hypersensitivity in chronic inflammatory and neuropathic pain.

**Keywords** Dorsal Root Ganglion; Thermosensation; Pain; Copine-6; Transient Receptor Potential Melastatin-3 (TRPM3)
**Subject Categories** Membranes & Trafficking; Neuroscience

## Introduction

Sensory nerves detect changes in external and internal environment and convey this information to the central nervous system (CNS). Somatosensory neurons are a heterogeneous population of neurons differentially attuned to respond to specific stimuli (e.g. mechanical, thermal, chemical) and within different stimulus strength (e.g. light touch vs. noxious pinch). This differential sensitivity arises from morphological features of different somatosensory fiber types and their different gene expression patterns. Increasing availability of single-cell transcriptomic approaches resulted in rapid development of understanding of genetic heterogeneity of somatosensory neurons, with 11 or more distinct classes reported (Jung et al, 2023; Li et al, 2016; Qi et al, 2024; Usoskin et al, 2015). Multiple databases listing patterns of gene expression of mammalian somatosensory neurons are now available (http://research-pub.gene.com/XSpeciesDRGAtlas/; https://mousespinal.brain-map.org/; https://painseq.shinyapps.io/publish/; https://rna-seq-browser.herokuapp.com/; Jung et al, 2023; Li et al, 2016). This wealth of single-cell transcriptomic data provides immense opportunity for 'data mining' and should be leveraged to develop better understanding of molecular mechanisms of somatosensation. To this end, we analyzed accessible transcriptomic data (Jung et al, 2023; Li et al, 2016) to identify genes that are highly expressed in sensory neurons but whose function is hitherto obscure, with the aim to determine their possible function in somatosensation. One of the genes that consistently appeared as highly expressed in several subpopulations of dorsal root ganglion (DRG) neurons is *Cpne6*, gene coding for Copine-6.

Copine-6, also known as N-Copine (Neuronal-Copine), is a calcium-dependent phospholipid-binding protein of the Copine family (Copine 1–9, coded for by *Cpne1-9* genes) (Creutz et al, 1998; Khvotchev and Soloviev, 2024; Nakayama et al, 1998). All Copines have two $Ca^{2+}$-binding C2 domains at the N-terminus and a von Willebrand factor A (vWA) domain at the C-terminus; Copines are the first intracellular proteins with a vWA domain to be described (Cowland et al, 2003; Creutz et al, 1998).

Copine-6 was initially thought to be brain-specific (Nakayama et al, 1999), and research to date has predominantly focused on its role in the CNS. Copine-6 was shown to impact kinetics of miniature excitatory post-synaptic currents (mEPSCs), affect synaptic vesicle recycling at contact sites, and alter hippocampal long-term potentiation (LTP) and depression (LTD), thereby influencing cognitive function through modulation of hippocampal dendritic spine structural plasticity (Burk et al, 2018; Chen et al, 2017; Reinhard et al, 2016). Mechanistically, Copine-6 was shown to translocate to the plasma membrane in response to intracellular calcium elevations, thus regulating membrane cycling of its interacting proteins (Ilacqua et al, 2018; Perestenko et al, 2015; Perestenko et al, 2010). Therefore, by acting as a postsynaptic $Ca^{2+}$

[1]Department of Pharmacology; The Key Laboratory of New Drug Pharmacology and Toxicology; The Key Laboratory of Neural and Vascular Biology, Ministry of Education; The Hebei Collaboration and Innovation Center for Mechanism, Diagnosis and Treatment of Neurological and Psychiatric Diseases, Hebei Medical University, Shijiazhuang, Hebei, China. [2]School of Biomedical Sciences; Faculty of Biological Sciences, University of Leeds, Leeds, UK. [3]Tongji Medical College, Huazhong University of Science and Technology, Wuhan, China. [4]These authors contributed equally: Yiting Gao, Shengxiang Yan.✉E-mail: gaohx686@hebmu.edu.cn; n.gamper@leeds.ac.uk

sensor, Copine-6 was shown to regulate membrane delivery of AMPA receptors during synaptic potentiation (Tan et al, 2023). Copine-6 is associated with conditions such as epilepsy and depression, with elevated expression in the temporal lobes of patients with temporal lobe epilepsy and reduced expression in the hippocampus of rat models of depression (Han et al, 2018; Zavala-Tecuapetla et al, 2020; Zhu et al, 2016). Despite high levels of expression in DRG neurons (Li et al, 2016), the potential role of this intriguing protein in the somatosensory system has not been explored.

Here we used genetic manipulations of Copine-6 expression in the DRG to reveal its function. We show that Copine-6 specifically enhances noxious heat sensitivity by partnering with transient receptor potential melastatin-3 (TRPM3) channel in a subpopulation of DRG neurons. Copine-6 physically interacts with TRPM3 and enhances channel function by promoting its translocation to the plasma membrane. These findings uncover a new role of Copine-6 in the peripheral nervous system and elucidates an essential cellular mechanism involved in thermosensation.

## Results

### Copine-6 is expressed mostly in medium and large-diameter DRG neurons

First, we used immunofluorescence to delineate the expression of Copine-6 across DRG neurons of different sizes. Typically, DRG neurons can be categorized by somatic diameter into: small ($< 25$ μm; predominantly C-fibers), medium (25–35 μm; predominantly Aδ-fibers), and large ($> 35$ μm; predominantly Aβ-fibers) (Middleton et al, 2022). Confocal images of DRG cryosections were segmented by Cellpose and analyzed using ImageJ (see "Methods"). Copine-6 was predominantly expressed in medium and large diameter DRG neurons (Fig. 1A) (25–35 μm: 2308/3701, 62.36%; >35 μm: 1599/2203, 72.58%), and less so in small diameter neurons ($< 25$ μm: 2445/6520, 37.50%). We then performed co-immunostaining of Copine-6 with either A-type fiber marker, neurofilament 200 (Hogea et al, 2021) (NF200; Fig. 1B), C-type fiber marker, peripherin (Fukazawa et al, 2023) (Fig. 1B) or satellite glial cell (SGC) marker, diazepam binding inhibitor (Li et al, 2024) (DBI; Fig. 1C). With regards to Copine-6 positive neurons, 62.7% (844/1346) were also labelled with NF200 and 18.0% (416/2314) were labelled with peripherin. There was no Copine-6 labelling in DBI-positive SGCs (Fig. 1D). Thus, Copine-6 is expressed in large proportions of Aδ and Aβ fibers, a smaller proportion of C fibers and is not detectable in SGCs.

### Loss of Copine-6 in the DRG selectively impairs noxious heat sensation

To investigate the possible role of Copine-6 in somatosensory transmission, we manipulated its expression in the DRG in vivo. First, we used in vivo gene silencing with shRNA. Four *Cpne6* shRNAs were screened in vitro (HEK293T cells and cultured DRG neurons; Fig. EV1A–D), and the most efficacious shRNA2 was packaged into AAV9 virions for subsequent experiments. AAV9-U6-Copine-6 shRNA2-EGFP or AAV9-U6-Scramble shRNA-EGFP

viral particles were locally injected into the L4 and L5 DRGs of rats (Fig. 2A). Knockdown efficiency is analyzed in Figs. 2B and EV1. Generally, both the *Cpne6* mRNA and Copine-6 protein levels were reduced by more than 50% in the DRG, while other copines expressed in the DRG (Copine-3, -4 and -8) were not significantly affected.

Four weeks after viral injection (to ensure full viral cargo expression), a series of behavioral tests was conducted (Fig. 2C–H). Strikingly, Copine-6 knockdown in the DRG significantly decreased threshold sensitivity to noxious heat without affecting other sensory modalities, such as mechanical and cold sensitivity, as well as proprioception. In the Hargreaves test for noxious heat sensitivity, the thermal withdrawal latencies for Scrambled shRNA and Copine-6 shRNA groups were $9.5 \pm 0.3$ s ($n = 9$) and $13.3 \pm 0.8$ s ($n = 11$, $P < 0.001$) respectively (Fig. 2E). There was no effect on mechanical sensitivity (Fig. 2C,D), cold sensitivity (Fig. 2F), balance (Fig. 2G), or gait (Fig. 2H).

Next, we tested the effect of Copine-6 DRG knockdown on heat hyperalgesia in the chronic inflammatory pain model (plantar paw injection of complete Freund's adjuvant, CFA, 50 μL). Knockdown of Copine-6 expression consistently reduced noxious heat sensitivity throughout the timeline of the experiment (Fig. 2I). Kinetics of development and relative magnitude of CFA-induced hyperalgesia were not affected, rather the entire sensitivity range was shifted towards higher temperatures. The latter suggests that Copine-6 is not involved in remodeling of heat sensitivity during chronic inflammation, rather, it is important for setting the threshold for the noxious range of heat sensitivity. Mechanical sensitivity or inflammation-induced mechanical hyperalgesia were unaffected (Fig. 2J). Similar results were obtained in the spared nerve injury (SNI) neuropathic pain model: Copine-6 knockdown reduced noxious heat sensitivity throughout the experiment, without an effect on mechanical sensitivity (Appendix Fig. S1A,B); the hypersensitivity in these experiments was not as pronounced as in CFA experiments but displayed an expected trend. The kinetics of hyperalgesia development was not affected but there was a significant recovery at day-7 after injury in the Copine-6 knockdown group, which was not seen in the Scrambled shRNA control group.

To confirm these initial findings, we utilized CRISPR/Cas9 technology to develop Copine-6 knockout mice. Specific guide RNAs (gRNAs) targeting *Cpne6* were employed to induce double-strand breaks before exon 2 and after exon 16, preventing the expression of *Cpne6* (Fig. 3A–C). The expression of Copine-6 was absent in the DRG of the *Cpne6* knockout ($Cpne6^{-/-}$) mice (Fig. 3C), while the expression of other *Cpne* family members was unchanged (Appendix Fig. S2A); *Cpne6, Cpne3* and *Cpne4* genes expressed at the highest level in the DRG (Appendix Fig. S2B).

$Cpne6^{-/-}$ is a global knockout, hence, control for the effects of Copine-6 in the DRG specifically, we also produced 'DRG-rescue' animals by injecting neuron-specific Copine-6 overexpression virions, AAV9-hSyn-Copine-6-EGFP (with an empty virus, AAV9-hSyn-EGFP, as a control) into the L3 and L4 DRGs of the $Cpne6^{-/-}$ mice (Fig. 3D,E). Of note, while in rats L4, L5, and L6 spinal nerves are major contributors to the sciatic nerve, in mice the contributing nerves are L3, L4, and L5 (Rigaud et al, 2008). The rest of Fig. 3 reports behavioral characterization of both $Cpne6^{-/-}$ and $Cpne6^{-/-}$-DRG-rescue mice with additional $Cpne6^{-/-}$ mice testing provided in Fig. EV2.

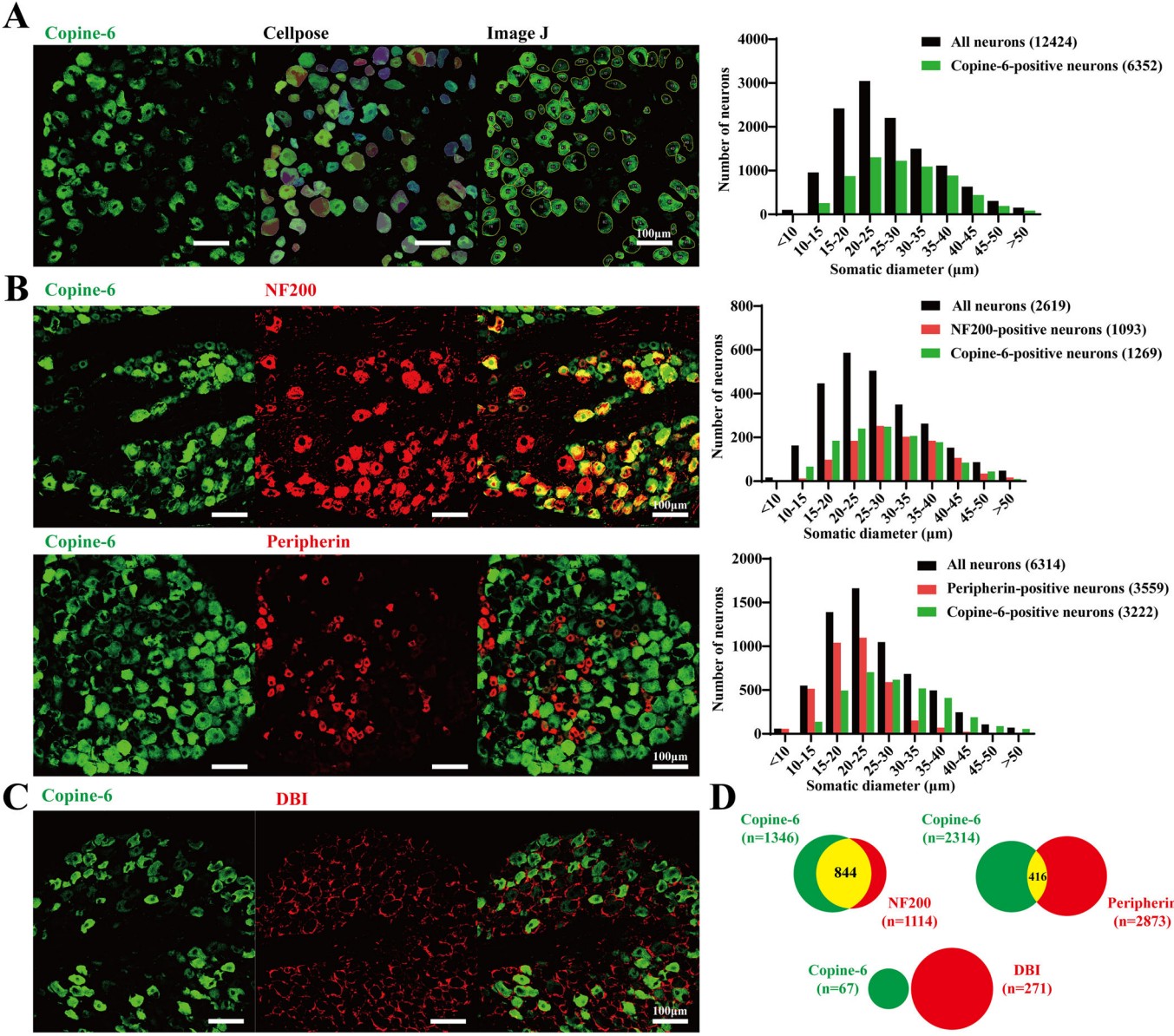

**Figure 1. Expression of Copine-6 in rat DRG.**

(A–C) Immunofluorescence analysis of rat DRG sections. Cell bodies of DRG neurons were auto-detected with Cellpose, and the masks obtained were loaded into ImageJ to acquire the area (shown in (A)). The somatic diameters of neurons were measured and plotted as size-distribution histograms on the right. Scale bars are 100 µm. (B) Co-immunostaining of DRG sections with antibodies against Copine-6 and NF200 (top) or peripherin (bottom). (C) Co-immunostaining of DRG sections with antibodies against Copine-6 and satellite glial cell (SGC) marker, DBI. (D) Pie chart quantify the proportions of DRG neurons expressing NF200, peripherin or DBI (red), and Copine-6 (green). Shown in yellow are the proportions of neurons co-expressing the NF200 or peripherin and Copine-6. Source data are available online for this figure.

$Cpne6^{-/-}$ mice exhibited normal mechanical sensitivity, cold sensitivity, balance, and gait in the von Frey (Figs. 3F and EV2A), dry ice (Figs. 3I and EV2B), rotarod (Fig. EV2C), and ink footprint (Fig. EV2D) tests, respectively. However, in both the Hargreaves (Figs. 3G and EV2E) and hot plate (Figs. 3H and EV2F,G) tests, $Cpne6^{-/-}$ mice exhibited significantly reduced heat sensitivity, compared to wild-type controls; these effects were seen in both male and female mice (Fig. EV2A,E–G). Importantly, this deficit was fully restored by the DRG-rescue. Specifically, the heat withdrawal latencies in the Hargreaves test were 8.8 ± 0.8 s

($n = 8$, $P < 0.001$) for the $Cpne6^{+/+}$ group, 13.8 ± 0.9 s ($n = 6$) for the $Cpne6^{-/-}$ group and 6.1 ± 0.5 s ($n = 6$, $P < 0.001$) for the $Cpne6^{-/-}$ DRG-rescue (Fig. 3G). The difference was even more pronounced in the perception of higher temperatures in the hot plate test, with heat withdrawal latencies for $Cpne6^{+/+}$ group ($n = 8$), $Cpne6^{-/-}$ group ($n = 6$) and $Cpne6^{-/-}$ DRG-rescue group ($n = 6$) as follows: at 45 °C: 35.0 ± 1.4 s, 35.0 ± 2.2 s, 32.5 ± 0.7 s; at 50 °C: 23.3 ± 1.1 s, 30.6 ± 1.7 s, 21.7 ± 1.7 s; at 52 °C: 15.3 ± 1.0 s, 21.4 ± 2.3 s, 14.3 ± 1.2 s; at 55 °C: 6.4 ± 0.6 s, 11.3 ± 1.7 s, 6.9 ± 0.4 s (Fig. 3G).

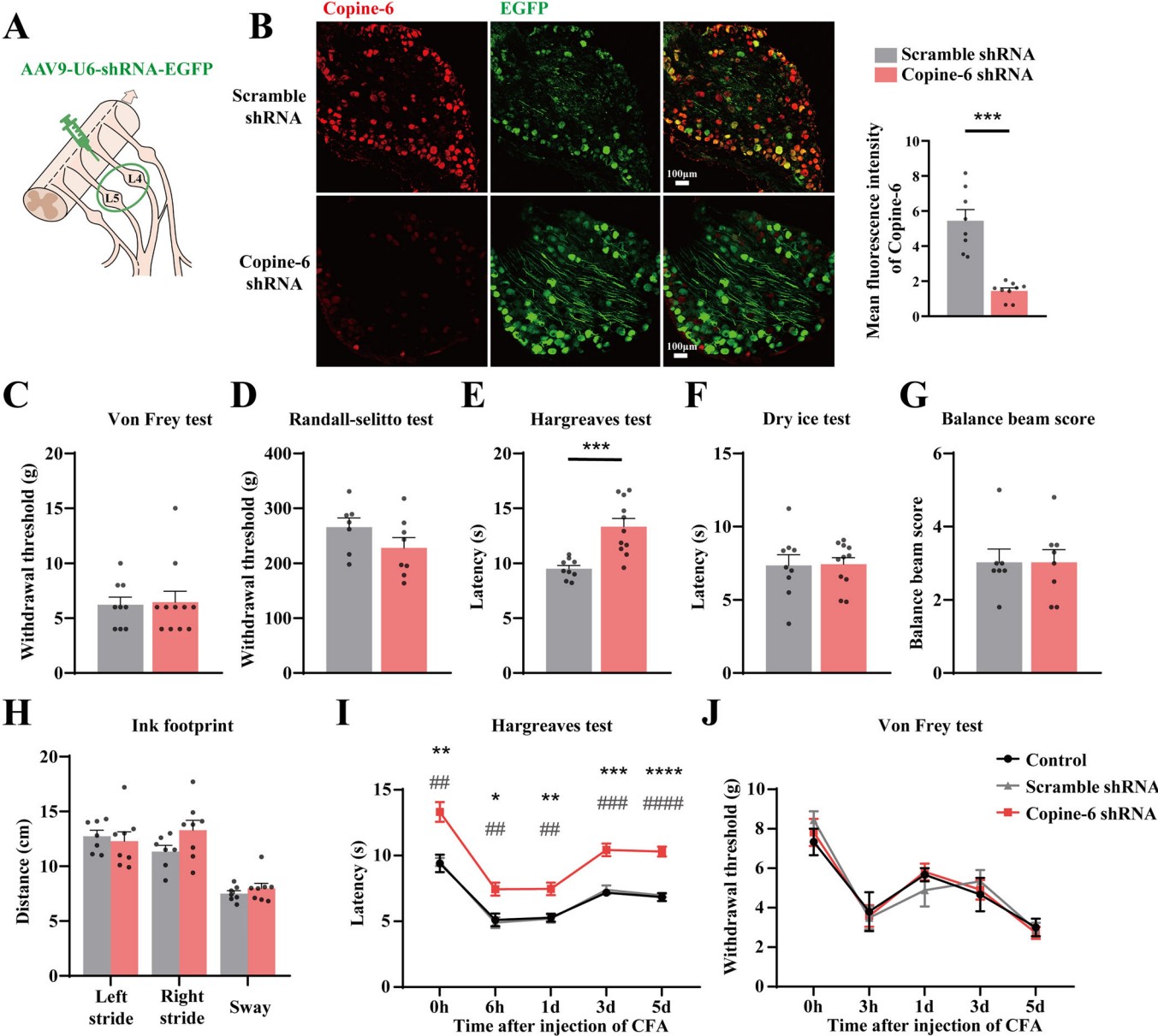

**Figure 2.** shRNA-mediated viral knockdown of Copine-6 in the DRG in vivo decreases thermal sensitivity in rats.

(A) Schematic of the local DRG injection of the viral constructs. (B) Confocal micrographs of Copine-6 immunoreactivity (red) and EGFP fluorescence (green) in the DRG sections of virus-infected animals (45 days after injection). Scale bars, 100 μm. Shown on the right is the quantification of Copine-6 knockdown efficiency ($n = 8$ sections from three animals in Scramble shRNA group and $n = 9$ sections from three animals in Copine-6 shRNA group. Data are shown as mean ± SEM. ***$P < 0.001$; two-tailed independent $t$ test. (C–H) Results of the behavioral tests performed on rats that received viral injections ($1 \times 10^{13}$ viral genomes/mL; 2 μL) of either Copine-6 shRNA or Scramble shRNA control virions. The tests were as follows: von Frey (C), Randall–Selitto (D), Hargreaves (E), dry ice (F), balance beam (G) and ink footprint (H) tests. For (C, E, F), $n = 9$ rats in Scramble shRNA group and $n = 11$ rats in Copine-6 shRNA group; for (D, G, H), $n = 7$ rats in Scramble shRNA group and $n = 8$ rats in Copine-6 shRNA group. Data are shown as mean ± SEM. (E) ***$P < 0.001$; two-tailed independent $t$ test. (I, J) On the 35th day after viral injection of either Copine-6 shRNA or Scramble shRNA control virions rats were injected with 50 μL of complete Freund's adjuvant (CFA) into the plantar surface of right hindpaw and Hargreaves (I) or von Frey (J) tests were performed at time intervals indicated. For (I, J), $n = 6$ rats in naïve control group, $n = 9$ rats in Scramble shRNA group and $n = 11$ rats in Copine-6 shRNA group. Data are shown as mean ± SEM. *Indicate difference from naïve control group and # indicate difference from Scramble shRNA group at time points indicated: 0 h, **$P = 0.0046$, ##$P = 0.0012$; 6 h, *$P = 0.0137$, ##$P = 0.0026$; 1 d, **$P = 0.0045$, ##$P = 0.0032$; 3 d, ***$P = 0.0001$, ###$P = 0.0002$; 5 d, ****$P < 0.0001$, ####$P < 0.0001$ (two-way ANOVA with Bonferroni multiple comparisons test). Source data are available online for this figure.

We also tested $Cpne6^{-/-}$ and $Cpne6^{-/-}$-DRG rescue mice in a chronic inflammatory pain model (CFA, 25 μL into the right plantar hind paw): results were consistent with the DRG-directed knockdown in rats. $Cpne6$ deletion significantly reduced noxious heat, but not mechanical sensitivity throughout the experiment (Figs. 3J,K and EV2H–L). As seen above, hyperalgesia was not prevented but it developed to higher thresholds, as compared to the WT mice. The effect of $Cpne6$ deletion was entirely reversed by the DRG rescue (Fig. 3J).

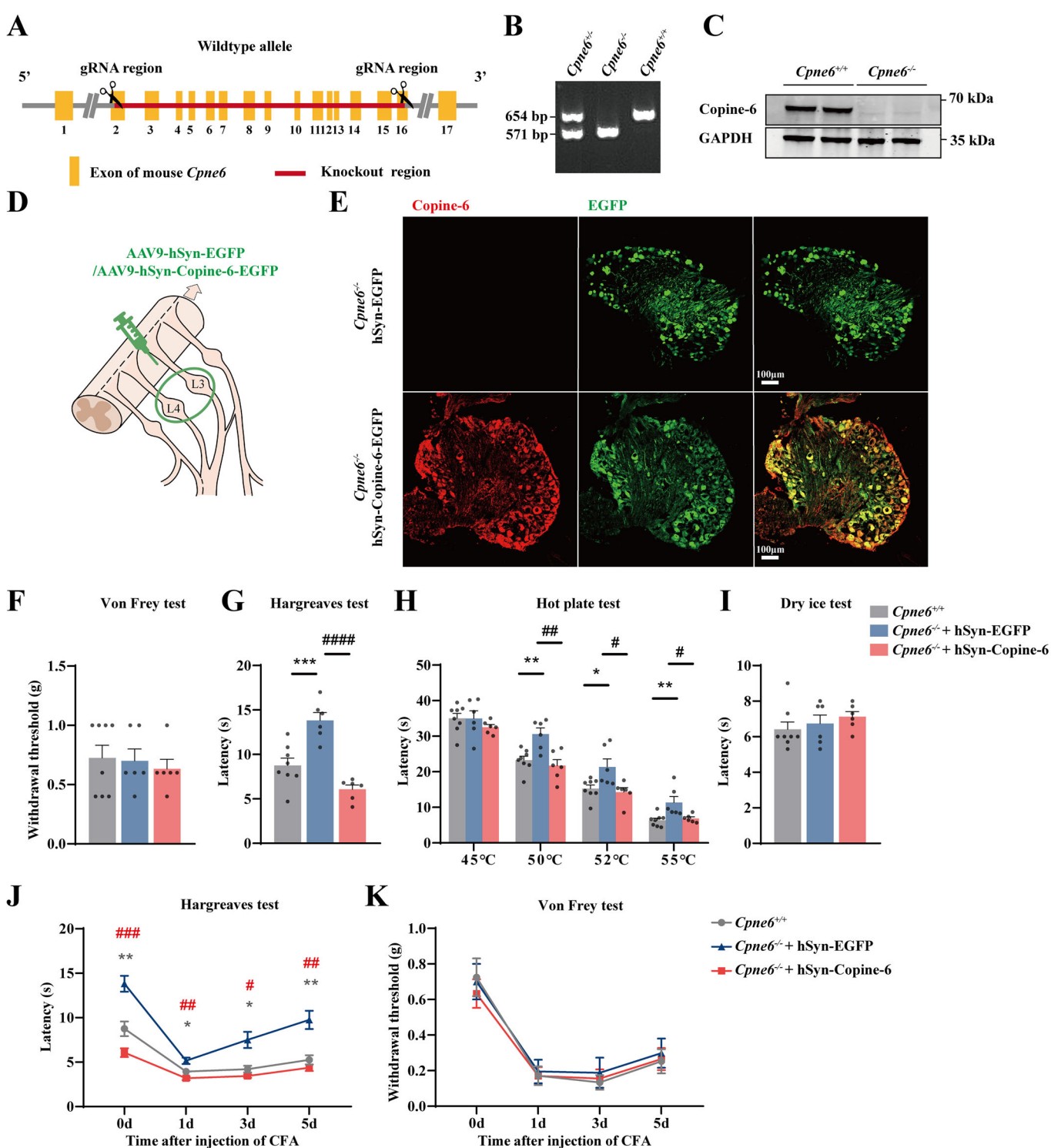

We also tested if *Cpne6* deletion affects itch (Fig. EV2M–O). No difference was seen in Cpne6⁻/⁻ mice compared to WT mice in various itch models such as histamine-independent chloroquine (CQ) acute itch model, histamine-dependent compound 48/80 acute itch model and diphenylcyclopropenone (DCP) chronic itch model.

## Deciphering molecular mechanism of Copine-6 involvement in thermosensation

Several molecular sensors were proposed to mediate noxious heat sensation in mammals but a recent series of studies from Voets and McNaughton laboratories rounded up four TRP channels that are

**Figure 3.   Global deletion and DRG-rescue of Copine-6 in mice induce reciprocal change in noxious heat sensitivity.**

(A) Overview of the targeting strategy for the construction of $Cpne6^{-/-}$ knockout mice. (B) Genotyping results showing $Cpne6^{+/-}$, $Cpne6^{-/-}$ and $Cpne6^{+/+}$ gene products. (C) Representative western blot analysis for Copine-6 expression in the DRG of $Cpne6^{+/+}$ and $Cpne6^{-/-}$ mice. (D) Schematic representation of local DRG-rescue by injection of the AAV9-hSyn-Copine-6-EGFP virions into the DRG of $Cpne6^{-/-}$ mice. (E) Confocal micrographs of Copine-6 immunofluorescence (red) and EGFP fluorescence (green) from the DRGs of $Cpne6^{-/-}$ mice DRG-injected with either AAV9-hSyn-Copine-6-EGFP virions or AAV9-hSyn-EGFP control virions (1 × 10$^{13}$ viral genomes/mL; 2 μL). Scale bars are 100 μm. (F-I) Results of the behavioral tests performed on of $Cpne6^{+/+}$ and $Cpne6^{-/-}$ mice DRG injected with either AAV9-hSyn-mCopine-6-EGFP virions or AAV9-hSyn-EGFP control virions. Tests are as follows: von Frey (F), Hargreaves (G), hot plate (H) and dry ice (I). $n = 8$ mice in $Cpne6^{+/+}$ group, $n = 6$ mice in both $Cpne6^{-/-}$ groups. Data are shown as mean ± SEM. *Indicate difference from $Cpne6^{+/+}$ group and # indicate difference in $Cpne6^{+/+}$-Copine-6 group. G, ***$P = 0.0007$; ####$P < 0.0001$. H, 50 °C, **$P = 0.0062$, ##$P = 0.0021$; 52 °C, *$P = 0.0292$, #$P = 0.0164$; 55 °C, **$P = 0.006$, #$P = 0.0232$ (one-way ANOVA followed by Bonferroni multiple comparisons test). (J, K) On the 35$^{th}$ day after viral injection of either AAV9-hSyn-Copine-6-EGFP virions or AAV9-hSyn-EGFP control virions mice were injected with 25 μL of CFA into the plantar surface of right hindpaw and Hargreaves (J) or von Frey (K) tests were performed at time intervals indicated; $n = 8$ mice in $Cpne6^{+/+}$ group, $n = 6$ mice in both $Cpne6^{-/-}$ groups. Data are shown as mean ± SEM. * indicate difference from $Cpne6^{+/+}$ group and # indicate difference from $Cpne6^{-/-}$ + hSyn-Copine-6 group at time points indicated. J, 0 d, **$P = 0.0029$, ***$P = 0.0001$; 1 d, *$P = 0.0372$, ##$P = 0.0029$; 3 d, *$P = 0.0252$, #$P = 0.0113$; 5 d, **$P = 0.009$, ##$P = 0.0045$. Two-way ANOVA with Bonferroni multiple comparisons test. Source data are available online for this figure.

necessary and sufficient for thermosensation: TRPV1, TRPA1, TRPM3 and TRPM2 (Tan and McNaughton, 2016; Vandewauw et al, 2018; Vilar et al, 2020; Vriens et al, 2014). First, using the DRG transcriptome atlas (Jung et al, 2023), we analyzed the distribution of $Cpne6$ and the four heat-related TRP channels across different subtypes of DRG neurons. The analysis revealed that the highest degree of overlap was between $Cpne6$ and $Trpm3$ (both in mouse and human DRGs), while co-expression with other thermosensitive TRP channels was much less pronounced (Fig. 4A,B). These data were broadly confirmed by immunofluorescence imaging of rat DRG sections labelled with antibodies against Copine-6 and TRPM3. Extensive co-localization of Copine-6 with TRPM3 was found, with 53.8% of TRPM3-positive neurons also expressing Copine-6 (Fig. 4C); specificity of Copine-6 and TRPM3 antibodies was confirmed in Appendix Fig. S3. There was significantly less co-localization of Copine-6 with TRPV1 (28.0%, $P < 0.0001$) and TRPA1 channels (25.6%, $P < 0.0001$) (Fig. 4D,E). The high degree of co-expression between Copine-6 and TRPM3 was confirmed with RNAscope (Appendix Fig. S4), with 60.9% of $Trpm3$ positive cells also labelled with $Cpne6$ probe.

TRPM2 immunofluorescence was generally weak and widely distributed in the DRG, thus, it was excluded from co-localization analysis with Copine-6. This was in agreement with generally lower but more ubiquitous $Trpm2$ mRNA abundance (Fig. 4A).

Next, we examined whether Copine-6 had the ability to regulate heat-related TRP channel activity in a heterologous system. We performed whole-cell patch-clamp recordings from Chinese Hamster Ovary (CHO) cells transiently expressing TRPV1, TRPA1, TRPM3 or TRPM2, alone or in combination with Copine-6 (GFP was co-transfected as a reporter) (Fig. 5). Channels were activated by their respective agonists; capsaicin (TRPV1, 1 μM), allyl-isothiocyanate (AITC; TRPA1, 100 μM), CIM0216 (TRPM3, 5 μM) or adenosine diphosphate ribose (ADPR; TRPM2, 25 μM). Co-expression of Copine-6 with TRPM3 significantly increased currents induced by CIM0216 (Fig. 5A,B). The CIM0216-induced current densities for the first and second CIM0216 application in CHO cells transfected with TRPM3 alone were 7.4 ± 0.6 pA/pF ($n = 32$) and 2.7 ± 0.3 pA/pF ($n = 18$), respectively; the corresponding current densities in cells co-transfected with Copine-6 and TRPM3 were 10.9 ± 0.8 pA/pF ($n = 37$, $P < 0.01$) and 4.5 ± 0.4 pA/pF ($n = 20$, $P < 0.01$), respectively (Fig. 5A,B). However, co-expression of Copine-6 did not alter currents induced by capsaicin (Fig. 5C,D), AITC (Fig. 5E,F) or ADPR (Fig. 5G,H), Additional electrophysiological characterization of the effect of Copine-6 on

TRPM3 is provided in Fig. EV3. In brief, presence of Copine-6 did not significantly affect such current properties as current-voltage relationships, rectification (Fig. EV3A–E) or concentration dependency for TRPM3 agonists, pregnenolone sulphate (Fig. EV3F) or CIM0216 (Fig. EV3G). Desensitization upon repeated agonist application was also not affected (Fig. 5A,B). Instead, presence of Copine-6 consistently increased current amplitudes by 50–90% (depending on the recording paradigm). The electrophysiological data presented above were obtained with γ3 splice variant of TRPM3, one of the four variants present in the DRG neurons, others are α2, α3 and γ2 (Uchida et al, 2019). To test if the Copine-6 effect is conserved amongst these variants, we co-expressed α2 and α3 variant with or without Copine-6 and in CHO cells and compared CIM0216-induced currents (Fig. EV3H). We found these are both similarly upregulated in the presence of Copine-6. We did not test γ2, but it is almost identical to γ3, with the exception of missing exon 15, which is also absent in α2. Hence, we believe that the three variants we tested cover all the sequence variability of the TRPM3 variants found in the DRG and all of them are sensitive to Copine-6 modulation.

Next, we examined the TRPM3 currents in dissociated DRG neurons (short-term culture) from the $Cpne6^{+/+}$ and $Cpne6^{-/-}$ mice using a gap-free recording, holding at -60 mV. Consistent with results obtained in heterologous expression system, CIM0216-induced currents were significantly smaller in $Cpne6^{-/-}$ mice, as compared to the WT controls. CIM0216-induced current density in DRG neurons from $Cpne6^{+/+}$ mice was 22.0 ± 3.3 pA/pF ($n = 27$) vs. 11.3 ± 2.2 pA/pF ($n = 17$; $P < 0.05$) in neurons form $Cpne6^{-/-}$ mice (Fig. 6A,B). Additionally, the proportion of DRG neurons exhibiting CIM0216-induced currents was significantly lower in $Cpne6^{-/-}$ mice, as compared to WT control: 27/37; 72.9% in $Cpne6^{+/+}$ vs. 17/35; 48.6% in $Cpne6^{-/-}$ mice ($P < 0.05$; Fig. 6C). In contrast, TRPV1 currents (induced by 1 μM capsaicin) and TRPA1 currents (induced by 100 μM AITC) were not different between in DRG neurons of $Cpne6^{+/+}$ and $Cpne6^{-/-}$ mice (Fig. EV4).

TRPM3 is a non-selective cation channel permeable to Ca$^{2+}$. We therefore further examined the function of TRPM3 channels by performing Ca$^{2+}$ imaging experiments on DRG neurons isolated from WT and $Cpne6^{-/-}$ mice. Consistently, the amplitude of the cytosolic calcium signal induced by CIM0216 in DRG neurons of $Cpne6^{-/-}$ mice was significantly lower than that in these from $Cpne6^{+/+}$ control mice (Fig. 6D). When classifying DRG neurons by cell diameter, the intracellular Ca$^{2+}$ signal induced by CIM0216 in DRG neurons < 25 μm (small) and 25–35 μm (medium) from

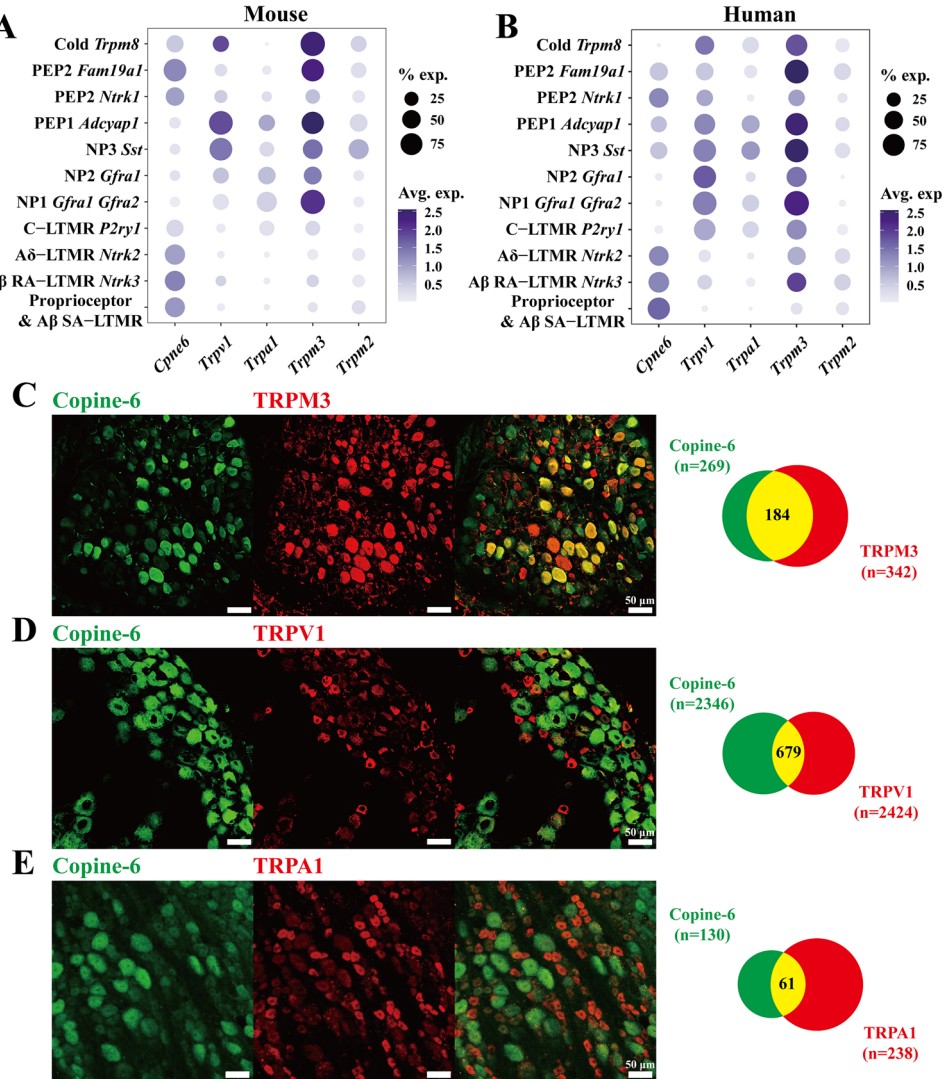

**Figure 4. Expression patterns of heat-sensitive ion channels TRPV1, TRPA1, TRPM3, TRPM2 and Copine-6 in the DRG.**

(A, B) Dot plots showing the proportions (dot size) and scaled mean expression levels (color) of genes of interest in sensory neuron subtypes of mouse (A) and human (B) DRG (data from Jung et al, 2023). (C–E) Confocal micrographs for co-immunostaining of DRG sections with antibodies against Copine-6 and either TRPM3 (C), TRPV1 (D) or TRPA1 (E). Scale bars are 50 µm. The pie charts on the right quantify the proportions of DRG neurons expressing Copine-6 (green) and either of the TRP channel (red). Shown in yellow are the proportions of neurons co-expressing Copine-6 and corresponding TRP channel. Source data are available online for this figure.

$Cpne6^{-/-}$ mice was significantly reduced (Fig. 6E). The proportion of medium-diameter (but not small or large) DRG neurons responding to CIM0216 with $Ca^{2+}$ transients was also significantly reduced, compared to WT controls (Fig. 6F). Similarly, the amplitudes of $Ca^{2+}$ transients induced by heat (45 °C) in DRG neurons of $Cpne6^{-/-}$ mice were also significantly lower than these in the $Cpne6^{+/+}$ mouse neurons (Fig. 6G,H). The proportion of DRG neurons responding to heat with $Ca^{2+}$ transients was also significantly reduced in $Cpne6^{-/-}$ mice, compared to $Cpne6^{+/+}$ controls (Fig. 6I).

To reveal potential other effects of Copine-6 that might have contributed to its effect on heat sensitivity, we compared general excitability parameters between DRG neurons from $Cpne6^{-/-}$ and $Cpne6^{+/+}$ control mice. We found that at room temperature, there was no significant difference in firing frequency (Fig. EV5A,B),

resting membrane potential (Fig. EV5C) or rheobase (Fig. EV5D) between DRG neurons from mice of either genotype.

We also tested the effects of $Cpne6$ knockout on nocifensive behaviour induced by the TRPM3 agonists CIM2016 and TRPV1 agonist capsaicin. The compounds were injected into the plantar paw (capsaicin: 10 µg/25 µL in 1.3% anhydrous ethanol; CIM0216: 10 nmol/25 µL in 10% DMSO) and the nocifensive behavior (incidence of licking or lifting the injected paw) was analyzed. $Cpne6$ knockout significantly reduced the nocifensive behavior induced by the CIM0216 but not capsaicin injection (Fig. 6J). Thus, in $Cpne6^{-/-}$ animals the CIM0216-induced nocifensive behavior was reduced to less than a half of that observed in WT controls ($Cpne6^{+/+}$ mice licked/lifted paws for 33.2 ± 5.8 s, $n = 6$, while $Cpne6^{-/-}$ mice licked/lifted paws for 16.7 ± 3.2 s, $n = 6$, $P < 0.05$). In contrast, the nocifensive behavior induced by capsaicin was not significantly affected.

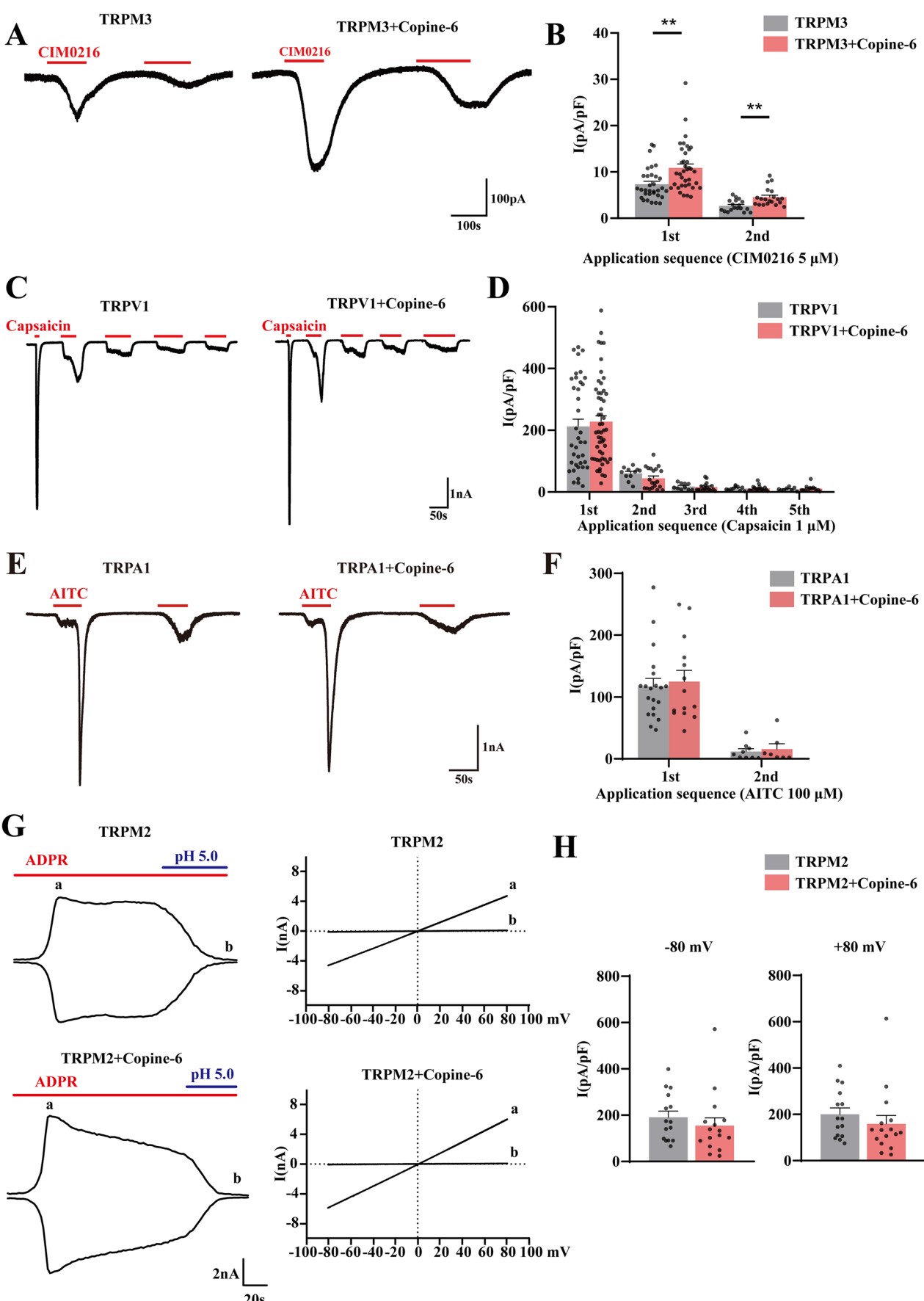

**Figure 5. Copine-6 co-expression selectively enhances TRPM3-mediated currents.**

Shown are the results of patch-clamp recordings from Chinese Hamster Ovary (CHO) cells transiently overexpressing TRPM3 (**A, B**), TRPV1 (**C, D**), TRPA1 (**E, F**) or TRPM2 (**G, H**), with or without Copine-6. (**A**) Representative currents evoked by 5 µM CIM0216 in CHO cells overexpressing TRPM3 with or without Copine-6. Recordings were performed with a gap-free protocol at a holding potential of −60 mV. (**B**) Comparison of peak TRPM3 current densities evoked by first and second application of CIM0216 (TRPM3 only: $n = 32, 18$; TRPM3 and Copine-6: $n = 37, 20$; 1st application, **$P = 0.001$, 2nd application, **$P = 0.001$; Mann–Whitney $U$ test). (**C, D**) Recordings similar to these shown in (**A, B**) from CHO cells overexpressing TRPV1 with or without Copine-6 with capsaicin (1 µM) used to activate TRPV1 currents in up to five repetitive applications. In (**D**) TRPV1 only: $n = 38, 11, 10, 10, 8$; TRPV1 and Copine-6: $n = 53, 20, 17, 17, 13$. (**E, F**) Recordings similar to these shown in (**A–D**) from CHO cells overexpressing TRPA1 with or without Copine-6 with allyl isothiocyanate (AITC, 100 µM) used to activate TRPA1 currents. In (**F**) TRPA1 only: $n = 20, 9$; TRPA1 and Copine-6: $n = 14, 7$. (**G**) Time course (left) and I-V relationships (right) of a whole-cell currents recorded from CHO cells overexpressing TRPM2 with or without Copine-6. Currents are recorded with adenosine diphosphate ribose (ADPR, 25 µM) in the intracellular solution upon breaking into the whole-cell configuration using a voltage ramp from −100 mV to +100 mV. Currents at −80 and +80 mV were analyzed. An acidified extracellular solution (pH 5.0) was applied at the end of the recording to inhibit TRPM2. (**H**) Comparison of peak current densities of ADPR-induced currents in CHO cells expressing TRPM2 ($n = 15$) or TRPM2 and Copine-6 ($n = 16$) at −80 and +80 mV. Source data are available online for this figure.

Importantly, i.p. injection of TRPM3 antagonist, isosakuranetin (Aloi et al, 2023; Straub et al, 2013) (2 mg/kg 15 min before testing) reduced heat sensitivity in the WT but not in the $Cpne6^{-/-}$ mice (Fig. 6K). Of note, the degree of increase in noxious heat withdrawal latency induced by TRPM3 antagonist and by the genetic deletion of $Cpne6$ was very similar. Combined, the data presented in this section strongly implicate functional coupling between Copine-6 and TRPM3 and suggest that reduction in noxious heat sensitivity observed after $Cpne6$ knockdown or knockout is mainly mediated by TRPM3.

## Copine-6 regulates TRPM3 plasma membrane abundance

Since Copine-6 only affected TRPM3 current density but not voltage dependence, desensitization or sensitivity to agonists, we hypothesized that Copine-6 may increase macroscopic TRPM3 currents by increasing TRPM3 abundance. To test this, we co-expressed TRPM3 with a C-terminal RFP tag (TRPM3-RFP) and Copine-6 carrying a C-terminal Flag-tag (Copine-6-Flag) in HEK293T cells. We then analyzed total and membrane protein fractions of TRPM3 by western blot. Interestingly, the membrane protein expression of TRPM3 was significantly increased by co-expression of Copine-6 (as compared to an empty vector control) without affecting the total TRPM3 levels (Fig. 7A–C). We further validated the impact of Copine-6 on TRPM3 plasma membrane expression using $Cpne6^{-/-}$ mice. qPCR results indicated that the absence of Copine-6 had no effect on $Trpm3$ mRNA expression in mouse DRG (Fig. 7D). Similarly, confocal microscopy images of immunofluorescence staining revealed no significant change in TRPM3 immunoreactivity in DRGs of $Cpne6^{-/-}$ mice (Fig. 7E). We then isolated plasma membrane proteins from the acutely extracted whole DRGs from both, $Cpne6^{-/-}$ and $Cpne6^{+/+}$ mice and assessed TRPM3 plasma membrane protein levels using western blot. Consistent with the results obtained in heterologous expression system, there was ~40% reduction ($P < 0.0001$) in TRPM3 membrane protein expression in the DRGs from $Cpne6^{-/-}$ mice, as compared to WT controls (Fig. 7F,G).

## Copine-6 facilitates TRPM3 trafficking and plasma membrane insertion

Copine-6, is a calcium-dependent phospholipid-binding protein, which is able to translocate to the cell membrane upon intracellular $Ca^{2+}$ elevation (Perestenko et al, 2010). To investigate its possible

role in promoting TRPM3 membrane trafficking, we stimulated cultured mouse DRG neurons with calcium ionophore ionomycin, TRPM3 channel agonist CIM0216, or noxious temperature (45 °C), and observed Copine-6 and TRPM3 membrane translocation using immunofluorescence and confocal microscopy (Fig. 8). In DRG neurons from $Cpne6^{+/+}$ mice, both Copine-6 and TRPM3 translocated to the plasma membrane after stimulation with ionomycin (5 µM, 10 min), CIM0216 (5 µM, 10 min), and heat (45 °C, 10 min) (Fig. 8A,E). Conversely, in DRG neurons from $Cpne6^{-/-}$ mice, TRPM3 failed to translocate to the plasma membrane in response to these stimuli, while Copine-6 immunofluorescence was absent (Fig. 8B,E). Moreover, employing the same approach, we tested if membrane translocation of Copine-6 affects TRPV1 and TRPA1 membrane trafficking in DRG neurons from the WT mice under stimulation with ionomycin (5 µM, 10 min), channel agonists (TRPV1: capsaicin, 1 µM, 10 min; TRPA1: AITC, 100 µM, 10 min), or high temperature (45 °C, 10 min). Interestingly, while these stimuli clearly induced plasma membrane translocation of Copine-6, the localization of TRPV1 and TRPA1 was unaffected (Fig. 8C,D,F,G). These experiments support escort function of Copine-6, which in DRG neurons is specific to TRPM3.

## Copine-6 physically interacts with TRPM3

A plasma membrane trafficking aid provided to TRPM3 by Copine-6 suggests a potential physical interaction between the two proteins. To investigate this, we tested whether TRPM3 and Copine-6 could be co-immunoprecipitated. HEK293T cells were transiently transfected with TRPM3-RFP alone, Copine-6-Flag alone, or both plasmids together. Anti-RFP and anti-Flag antibodies were used to detect TRPM3 and Copine-6 in the transfected cell lysates. Interestingly, Copine-6-Flag was specifically detected when co-transfected cell lysates were immunoprecipitated with an anti-RFP antibody. Similarly, TRPM3-RFP was successfully identified when cell lysates were immunoprecipitated with an anti-Flag antibody (Fig. 9A). Additionally, a proximity ligation assay (PLA) was conducted in DRG neurons to further confirm the physical interaction between TRPM3 and Copine-6 (Fig. 9B). This 'in situ proteomic' method tests for proximity between two proteins of interest with specific fluorescent puncta being developed only if two proteins are less than ~40 nm apart (Söderberg et al, 2006; Weibrecht et al, 2010) (see "Methods"). Indeed, we observed clear PLA signal in many cultured rat DRG neurons. In contrast, PLA between Copine-6 and TRPV1 returned no or only singular puncta in most of the neurons. The mean number of puncta per positive

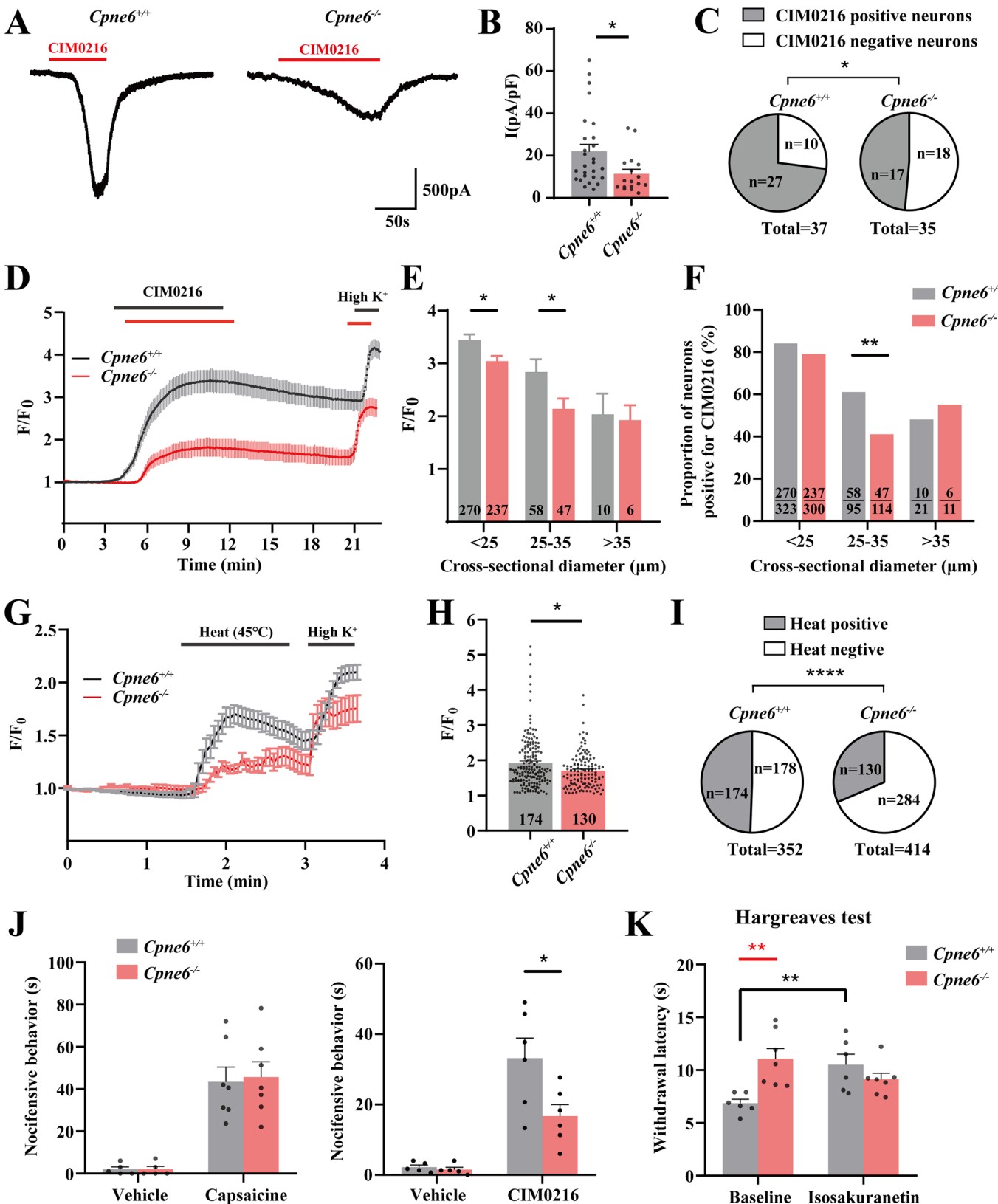

◄ **Figure 6.  Copine-6 deletion impairs native TRPM3 channel function in the DRG.**

(A) Example currents evoked by CIM0216 (5 µM) in DRG neurons from $Cpne6^{+/+}$ (left) or $Cpne6^{-/-}$ (right) mice. Recordings were performed with a gap-free protocol at a holding potential of −60 mV. (B) Comparison of peak CIM0216-evoked current densities in recordings as these shown in (A) from the DRG neurons from $Cpne6^{+/+}$ ($n = 27$) and $Cpne6^{-/-}$ ($n = 17$) mice; recordings from at least three independent preparations. Data are shown as mean ± SEM; *$P = 0.012$ (Mann–Whitney $U$ test). (C) Proportion of CIM0216-responding DRG neurons in mice of the indicated genotypes. *$P = 0.034$ (Pearson chi-square test). (D) Fluorescence $Ca^{2+}$ imaging of Fluo-4AM-loaded DRG neurons from $Cpne6^{+/+}$ (grey) and $Cpne6^{-/-}$ (pink) mice. CIM0216 was applied during periods indicated by the horizontal bars, neurons were identified by responses to extracellular solution containing 60 mM KCl ('High K$^+$') applied at the end of the recording. Shown in this panel are averages of all neurons from one field of view ($Cpne6^{+/+}$ group: $n = 47$ neurons; $Cpne6^{-/-}$ group: $n = 23$ neurons). (E) Comparison of calcium signal amplitude evoked by CIM0216 in DRG neurons of different sizes in two groups of mice ($Cpne6^{+/+}$: < 25, $n = 270$; 25–35, $n = 58$; > 35, $n = 10$; $Cpne6^{-/-}$ group: < 25, $n = 237$; 25–35, $n = 47$; $n > 35$, $n = 6$); recordings from at least three independent preparations; Data are shown as mean ± SEM; < 25, *$P = 0.013$; 25–35, *$P = 0.032$ (Mann–Whitney $U$ test). (F) Percentages of CIM0216-responding DRG neurons with different diameters in mice of the indicated genotypes. **$P = 0.004$ (Pearson chi-square test). (G) Fluorescence $Ca^{2+}$ imaging of Fluo-4AM-loaded DRG neurons from $Cpne6^{+/+}$ (grey) and $Cpne6^{-/-}$ (pink) mice. The perfusion chamber was heated to 45 °C during the time indicated by the horizontal bar; neurons were identified by responses to extracellular solution containing 60 mM KCl ('High K$^+$') applied at the end of the recording. Shown in this panel are averages of all neurons from one field of view ($Cpne6^{+/+}$ group: $n = 33$ neurons; $Cpne6^{-/-}$ group: $n = 21$ neurons). (H) Comparison of calcium signal amplitude evoked by heat in DRG neurons from mice of either genotype ($Cpne6^{+/+}$ group: $n = 174$; $Cpne6^{-/-}$ group: $n = 130$); recordings from at least three independent preparations. Data are shown as mean ± SEM; *$P = 0.019$ (Mann–Whitney $U$ test). (I) Proportion of heat-responding DRG neurons in mice of the indicated genotypes. ****$P < 0.0001$ (Pearson Chi-square test). (J) Nocifensive behavior (total time of licks or lifts of the paw) within 2 min after the plantar injection of capsaicin (10 µg/25 µL/paw; left) or CIM0216 (10 nmol/25 µL/paw; right), as compared to vehicle (1.3% alcohol or 10% DMSO in saline, respectively); a comparison between $Cpne6^{+/+}$ and $Cpne6^{-/-}$ mice; $n = 7$ mice for capsaicin group and $n = 6$ mice for CIM0216 group; $n = 5$ mice in both vehicle groups. Data are shown as mean ± SEM. Right, *$P = 0.031$ (two-tailed independent $t$ test). (K) Paw withdrawal latencies to thermal stimulation 15 min after the injection of 2 mg/kg isosakuranetin (i.p.) in $Cpne6^{+/+}$ and $Cpne6^{-/-}$ mice. $n = 6$ mice in $Cpne6^{+/+}$ group and $n = 7$ mice in $Cpne6^{-/-}$ group. Data are shown as mean ± SEM. Red * indicate the difference between $Cpne6^{+/+}$ and $Cpne6^{-/-}$ mice of the baseline group; **$P = 0.0022$. Black * indicate the difference between baseline group and isosakuranetin group of the $Cpne6^{+/+}$ mice; **$P = 0.0062$. Two-way ANOVA with Bonferroni multiple comparisons test. Source data are available online for this figure.

neuron for the Copine-6/TRPM3 pair was 31.4 ± 6.8 ($n = 22$ out of 60; $N = 3$), while for Copine-6/TRPV1 pair it was 5.4 ± 1.2 ($n = 17$ out of 60; $N = 3$; $P < 0.001$). These results collectively demonstrated that TRPM3 indeed interacts with Copine-6 physically.

To delineate the interaction domain(s) within Copine-6, which may be involved in the interactions with TRPM3, a series of truncated constructs of Copine-6 was designed and these were co-expressed with TRPM3-RFP and lysates were immunoprecipitated with antibodies to RFP. We made constructs expressing peptides corresponding to C2A (Copine-6$_{1-151}$), C2B (Copine-6$_{134-269}$) and vWA (Copine-6$_{269-557}$) domains, as well C2A-C2B tandem (Copine-6$_{1-269}$) and C2B-vWA tandem (Copine-6$_{134-557}$) (Fig. 9C). These mapping studies demonstrated that the first C2 domain (C2A) and vWA domain of Copine-6 were both necessary and sufficient for TRPM3 immunoprecipitation (Fig. 9D). The results were further corroborated with electrophysiology. The Copine-6 truncation constructs were co-expressed with TRPM3 in CHO cells and the effect on CIM0216-induced current density was investigated. These experiments revealed that only the vWA domain of Copine-6, along with the full-length Copine-6, could augment the TRPM3 current density. Measured at -80 mV TRPM3 current densities were as follows. TRPM3 alone: 34.1 ± 2.3 pA/pF ($n = 14$); TRPM3 + C2A: 30.3 ± 3.5 pA/pF ($n = 15$), TRPM3 + C2B: 37.7 ± 3.7 pA/pF ($n = 14$); TRPM3+vWA: 63.0 ± 8.5 pA/pF ($n = 17$) and TRPM3+full-lenth Copine-6: 57.0 ± 4.5 pA/pF ($n = 23$), respectively. Measured at +80 mV, TRPM3 current densities were as follows. TRPM3 alone: 139.5 ± 11.7 pA/pF ($n = 14$); TRPM3 + C2A: 113.9 ± 11.4 pA/pF ($n = 15$), TRPM3 + C2B: 155.4 ± 17.4 pA/pF ($n = 14$); TRPM3+vWA: 231.9 ± 19.4 pA/pF ($n = 17$) and TRPM3+full-lenth Copine-6: 208.6 ± 13.6 pA/pF ($n = 23$), respectively (Fig. 9E). These experiments revealed the pivotal role of the vWA domain of Copine-6 in its functional interaction with TRPM3. In conclusion, experiments in this series demonstrate that Copine-6 facilitates the translocation of TRPM3 to the plasma membrane, resulting in increased membrane protein abundance and enhanced TRPM3 currents. The effect is selective to

TRPM3 over other heat-sensitive TRP channels. C2A and vWA domains of Copine-6 participate in interaction with TRPM3, although only vWA domain appears necessary and sufficient for the upregulation of functional TRPM3.

Overall, Copine-6 is likely physically interacting with TRPM3 and escorting the channel to the membrane, thus, functioning to enhance the activity of TRPM3.

## Discussion

In the work that led up to the current study, we mined available RNA-seq datasets to identify genes that are highly expressed in somatosensory neurons but have unknown functionality in the somatosensory system. Our search criteria were very simple: (1) significant expression of the gene in the DRG across multiple published datasets; (2) limited or absent understanding of the corresponding protein's role in the somatosensory system. One of the genes that satisfied these criteria was *Cpne6*. Indeed it consistently appears as highly expressed in a subpopulation of peripheral somatosensory neurons in multiple single-cell RNA-seq datasets (http://research-pub.gene.com/XSpeciesDRGAtlas/; https://painseq.shinyapps.io/publish/; https://rna-seq-browser.herokuapp.com/; Jung et al, 2023; Li et al, 2016), as well as in Allen Spinal Cord Atlas. Yet, to the best of our knowledge, there is no reports on the possible role of this gene/protein in somatosensation. Originally, Copine-6 was thought to be specifically expressed in the brain (Nakayama et al, 1999), hence, research on Copine-6 has remained focused mostly on its role in the CNS.

Here, using global genetic deletion (along with DRG-rescue) of *Cpne6* in mouse and DRG-specific knockdown of *Cpne6* in rats we show that Copine-6 is specifically involved in regulation of sensitivity to noxious heat, having no significant effects on other sensory modalities, such as mechanosensitivity, sensitivity to cold or proprioception. Deletion or downregulation of Copine-6 markedly reduces noxious heat detection thresholds and reduces

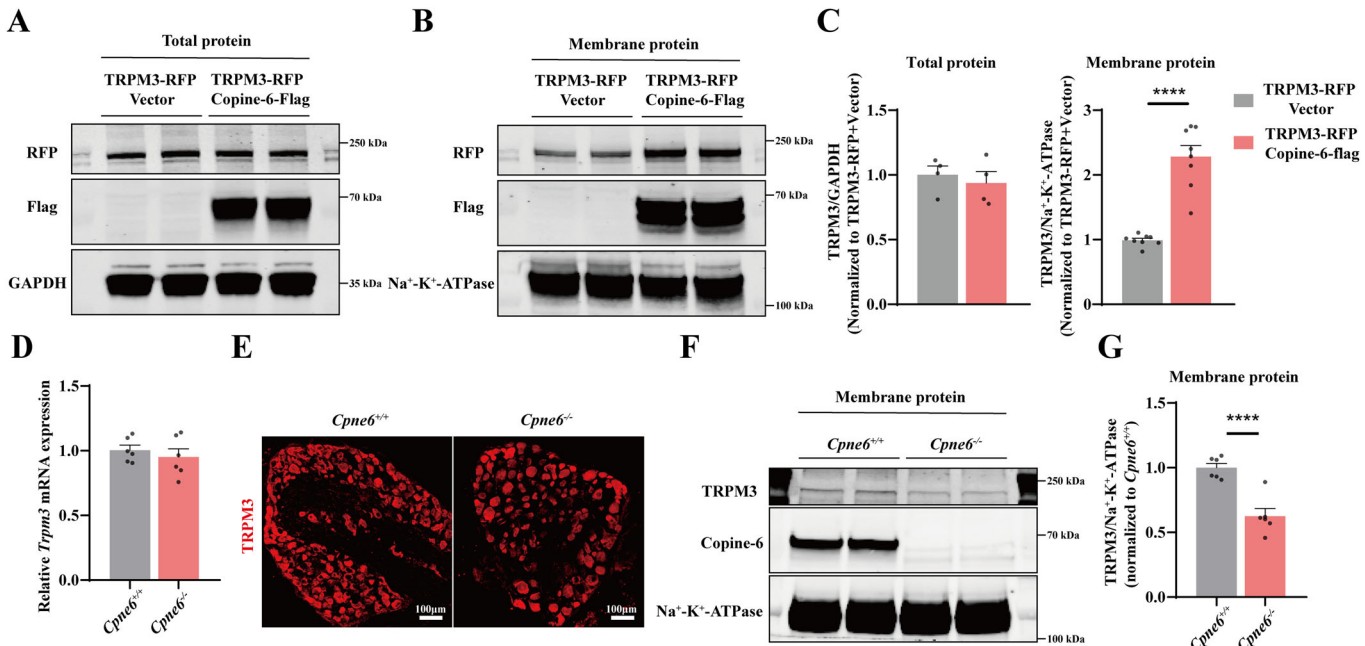

**Figure 7. Copine-6 increases TRPM3 plasma membrane protein abundance.**

(A) Representative western blot analysis of TRPM3 expression in whole-cell protein extracts from HEK293T transiently overexpressing RFP-tagged TRPM3 with or without Flag-tagged Copine-6. GAPDH was used as loading control. (B) Representative western blot analysis of TRPM3 in the membrane protein fractions from HEK293T cells transfected as in (A). $Na^+$-$K^+$-ATPase was used as a loading control. (C) Quantitative analysis of the TRPM3 band intensities from experiments shown in (A, B), normalized to GAPDH or $Na^+$-$K^+$-ATPase, respectively, and expressed as a ratio to the corresponding band in the TRPM3-RFP+Vector group. Total protein, $n = 4$. Membrane protein, $n = 8$. Data are shown as mean ± SEM. ****$P < 0.0001$ (two-tailed independent $t$ test). (D) $Trpm3$ mRNA levels in the DRG of $Cpne6^{-/-}$ mice compared to $Cpne6^{+/+}$ mice. $n = 6$. (E) Confocal micrographs of TRPM3 immunofluorescence in DRG sections from $Cpne6^{+/+}$ or $Cpne6^{-/-}$ mice. Scale bars are 100 μm. (F) Representative western blot analysis of TRPM3 expression in the membrane protein fractions from the DRG of $Cpne6^{+/+}$ or $Cpne6^{-/-}$ mice. (G) Quantitative analysis of the TRPM3 band intensities normalized to $Na^+$-$K^+$-ATPase in the plasma membrane, and then further normalized to the $Cpne6^{+/+}$ group. $n = 6$. Data are shown as mean ± SEM. ****$P < 0.0001$ (two-tailed independent t-test). Source data are available online for this figure.

heat hypersensitivity in chronic neuropathic and inflammatory pain models. By testing the effect of Copine-6 on four noxious heat sensors: TRPV1, TRPA1, TRPM3 and TRPM2 (Tan and McNaughton, 2016; Vandewauw et al, 2018; Vilar et al, 2020; Vriens et al, 2014), we identified TRPM3 as a sole target of Copine-6. Our data are consistent with a model, whereby Copine-6 binds to TRPM3 via its vWA domain and promotes insertion of the channel into the plasma membrane, with the process facilitated by intracellular $Ca^{2+}$.

We demonstrate that Copine-6 displays higher expression in medium- to large-diameter DRG neurons with relatively lower expression in small-diameter DRG neurons, this pattern is broadly consistent with RNA-seq databases (Jager et al, 2020; Jung et al, 2023; Li et al, 2016). We speculate, however, that it is the subpopulations of TRPM3-expressing peptidergic C-fibers and some TRPM3-expressing Aδ fibers that are important for Copine-6 effects on thermosensation (Fig. 4A,B). Furthermore, we show that Copine-6 is specifically expressed in neurons within the DRG and is not detectable in SGCs (Fig. 1C).

TRPM3 is a non-selective cation channel contributing to the detection of noxious heat in mammals (Vandewauw et al, 2018; Vriens et al, 2011). Accordingly, in hot plate tests, TRPM3 knockout mice display prolonged withdrawal latencies to temperatures above 52 °C, while sensitivity to cold and mechanical stimuli remain unchanged (Vriens et al, 2011). Furthermore, TRPM3 does

not mediate itch induced by endogenous pruritogens (Kelemen et al, 2021). This pattern of somatosensory functionality matches to what we observed with $Cpne6$ knockout or knockdown. Thus, both, $Cpne6^{-/-}$ mice and rat DRG-injected with $Cpne6$-targeting shRNA displayed reduced sensitivity to noxious heat, but their sensitivity to warm temperatures, cold, mechanical stimuli, as well as proprioception remained largely unchanged (Figs. 2 and 3). We also found that the deletion of $Cpne6$ did not affect acute itch (both histamine-dependent and histamine-independent) or chronic itch (Fig. EV2). Importantly though, viral overexpression of Copine-6 in the L3 and L4 DRG of $Cpne6^{-/-}$ mice fully rescued the impairment in heat sensation, indicating that Copine-6 in DRG neurons, rather than elsewhere in the somatosensory pathways, played a critical role in regulation of noxious heat sensitivity.

The membrane trafficking of sensory TRP channels is a crucial mechanism for tuning sensory neuron input/output relationships (Linley et al, 2010; Rivera et al, 2024) and, hence, the overall sensitivity of a nerve. Abnormal TRP channel trafficking is observed in many chronic pain conditions (Ferrandiz-Huertas et al, 2014). However, the mechanism underlying TRPM3 membrane trafficking remains largely unknown. A recent study has shown that activation of transcription factor 4 (ATF4) mediates TRPM3 membrane transport in DRG neurons by interacting with the motor protein KIF17, thereby enhancing heat sensitivity (Xie et al, 2021). Here we identified a novel and distinct trafficking

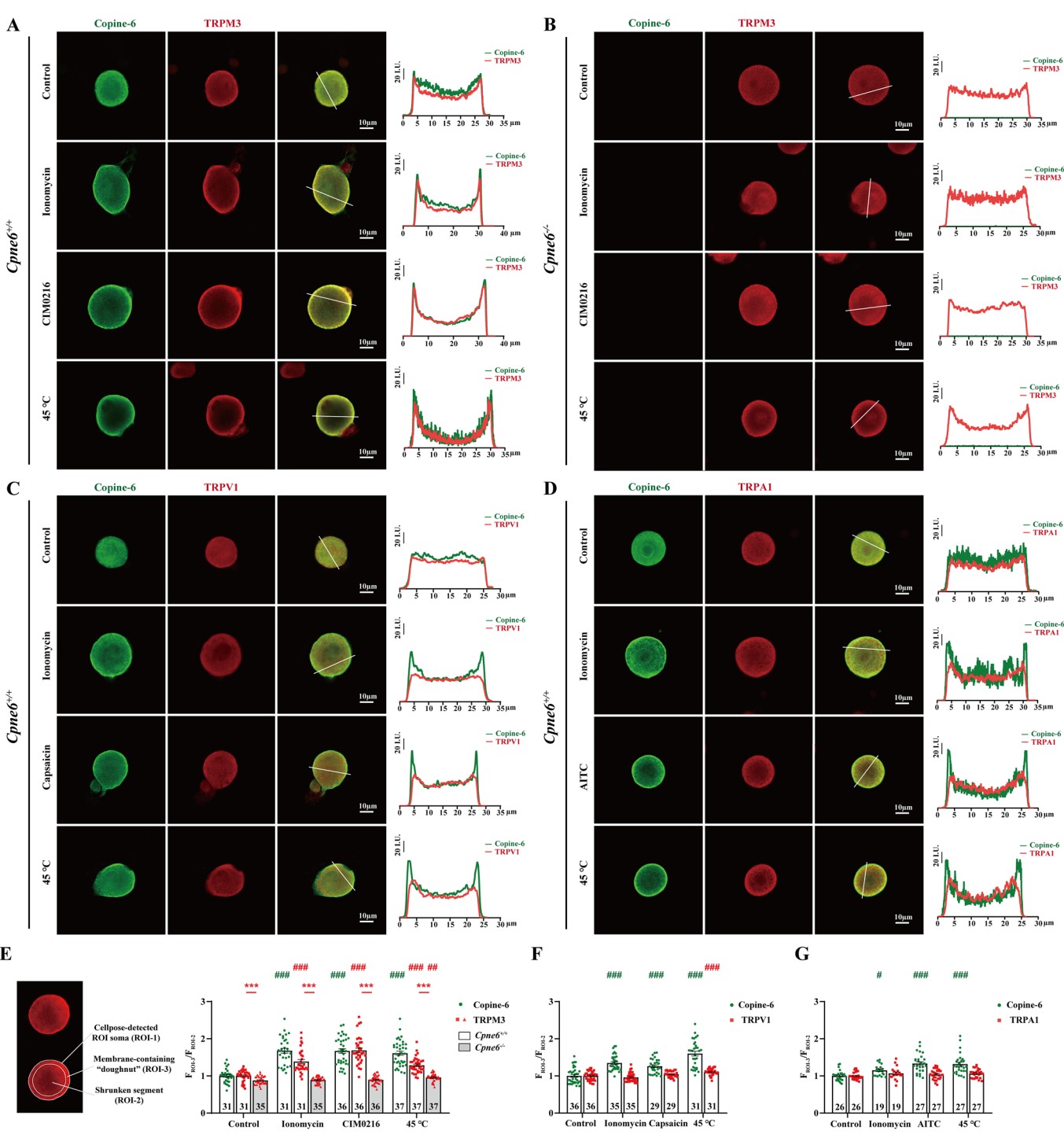

mechanism, whereby Ca²⁺-sensitive Copine-6 binds to TRPM3 and escorts it to the plasma membrane, resulting in increased TRPM3 membrane abundance, while having no such effect on other noxious heat-sensitive TRP channels tested.

We verified the interaction between Copine-6 and TRPM3 by immunoprecipitation performed in expression systems and by PLA performed on rat DRG neurons (Fig. 9A,B). Further structural analysis suggests that the C2A domain and the vWA domain of Copine-6 are potential regions for TRPM3 interactions, although

electrophysiological data indicate that only the vWA domain is necessary and sufficient for functional upregulation of TRPM3 currents (Fig. 9D,E). The vWA domains typically mediate protein-protein interactions (Tomsig and Creutz, 2002; Tomsig et al, 2003), but the function of the vWA domain of Copine-6 has not been well understood. Therefore, our study unveils one of such possible functions.

Previous studies suggested that the C2B domain of Copine-6 is responsible for Ca²⁺ binding and plasma membrane targeting

**Figure 8. Copine-6 promotes translocation of TRPM3 to the plasma membrane.**

(A–D) Confocal micrographs of the cultured, fixed DRG neurons from $Cpne6^{+/+}$ or $Cpne6^{-/-}$ mice (as indicated) co-immunolabelled for Copine-6 (green) and TRPM3 (red, **A, B**) or TRPV1 (red, **C**) or TRPA1 (red, **D**). Before fixation cells were treated with ionomycin (5 μM, 10 min), CIM0216 (5 μM, 10 min), capsaicin (1 μM, 10 min), AITC (100 μM, 10 min) or 45 °C (10 min), as indicated. Scale bars are 10 μm. Fluorescence intensity line scan profiles taken at the lines indicated for each cell are shown in the inserts on the right. (**E–G**) Analysis of the membrane localization of Copine-6 and TRPM3 (**E**), TRPV1 (**F**) and TRPA1 (**G**). Cell bodies of the DRG neurons were auto-detected with Cellpose, and the automatically detected segments were radially shrunk to 80%. Fluorescence of the cell membrane was analyzed in the "doughnut" space between the contracted and automatically detected segments (schematized in the inset; see "Methods"). (**E**) $N = 5$ mice in each group, (**F, G**) $N = 4$ mice in each group. Data are shown as mean ± SEM. (**E**) Red * indicate difference in TRPM3 ($F_{ROI-3}/F_{ROI-2}$) between $Cpne6^{+/+}$ and $Cpne6^{-/-}$ groups; \*\*\*$P < 0.001$. Two-tailed independent $t$ test. (**E–G**) Green # indicate difference from control group in Copine-6 ($F_{ROI-3}/F_{ROI-2}$). Red # indicate difference from control group in TRPM3 or TRPV1 ($F_{ROI-3}/F_{ROI-2}$). (**E**) Green ###$P < 0.001$; Red ###$P < 0.001$; Red ##$P = 0.004$. (**F**) Green ###$P < 0.001$ Red ###$P < 0.001$. (**G**) Green #$P = 0.02$; Green ###$P < 0.001$. One-way ANOVA with Bonferroni multiple comparisons test. Source data are available online for this figure.

(Perestenko et al, 2015; Tan et al, 2023). This raises a question: even if vWA domain of Copine-6 can bind to TRPM3, how can it promote membrane insertion without a C2B domain? Intriguingly, the vWA domain at the N-terminus of calcium-activated chloride channel regulator 1 (CLCA1) was shown to stabilize TMEM16A, a $Ca^{2+}$ activated $Cl^-$ channel (CaCC), on the cell surface by preventing its reinternalization. In turn, this increases TMEM16A abundance at the cell surface and results in augmentation of CaCC currents (Berry and Brett, 2020; Sala-Rabanal et al, 2017; Sala-Rabanal et al, 2015). Therefore, we hypothesize that the vWA domain of Copine-6 may play a similar role by increasing TRPM3 plasma membrane abundance. In this scenario, overexpression of vWA domain of Copine-6 'locks' more TRPM3 channels at the plasma membrane, resulting in larger currents in the expression system. In the case of full-length Copine-6, it would be expected that $Ca^{2+}$-induced TRPM3 membrane targeting is provided by C2 domains and membrane retention is bestowed by the vWA domain. This intriguing hypothesis requires further experimental validation.

Intracellular $Ca^{2+}$ elevations induced by various means, including (i) nonspecific elevation with membrane ionophore, ionomycin, (ii) activation of TRPV1, TRPA1 or TRPM3 with their specific ligands, or (iii) with 45 °C heat, invariantly induced translocation of Copine-6 to the plasma membrane of DRG neurons (Fig. 8). Consistently, ionomycin, noxious heat and CIM0216 induced Copine-6-dependent membrane insertion of TRPM3 (Fig. 8A,B). Hence, Copine-6 appears to function as a universal activity-dependent TRPM3 escort. We further speculate that this mode of action may be important to ensure TRPM3 membrane abundance (and, consequently, thermal sensitivity) is maintained throughout peripheral nerve activity, perhaps to offset channel desensitization.

Interestingly, both Copine-6 and TRPM3 are highly expressed in several brain regions, including hippocampus and olfactory bulb (https://mouse.brain-map.org/). TRPM3 gain-of-function mutations were associated with cognitive deficits and epilepsy (Burglen et al, 2023; Zhao et al, 2020). Hence, activity-dependent trafficking of this channel by Copine-6 in the brain may be of relevance to regulation of higher brain functions and warrant further investigation.

In summary, this study reveals a novel role for Copine-6 in peripheral somatosensory system: it functions as an escort protein facilitating insertion of TRPM3 into the plasma membrane. This, in turn, sets the noxious heat sensitivity range of corresponding somatosensory neurons. The ultimate physiological effect of Copine-6 deficiency is reduction in heat sensitivity within the noxious range and lessening of chronic pain-associated heat hyperalgesia. Our findings provide experimental and theoretical foundations for better understanding the physiology of thermal pain mechanisms.

# Methods

### Reagents and tools table

| Reagent/resource | Reference or source | Identifier or catalog number |
|---|---|---|
| **Experimental models** | | |
| HEK-293T cells (H. sapiens) | KCB | Cat#KCB200744YJ |
| CHO cells (C. griseus) | KCB | Cat#KCB83004YJ |
| C57BL6/J (M. musculus) | Charles River Laboratories | N/A |
| SD Rat (R. norvegicus) | Charles River Laboratories | N/A |
| $Cpne6^{+/-}$ mice (M. musculus) | cyagen | N/A |
| **Recombinant DNA** | | |
| rCpne6 pcDNA3.1-3xFlag-C | YouBio | N/A |
| pcDNA3.1-3xFlag-C | YouBio | N/A |
| mTrpm3 pcDNA3.1-T2A-TagRFP (NM_177341.4/γ3) | YouBio | N/A |
| mTrpm3 pcDNA3.1-TagRFP-C (NM_177341.4/γ3) | YouBio | N/A |
| mTrpm3 pcDNA3.1-T2A-TagRFP (NM_001035242.1/α2) | YouBio | N/A |
| mTrpm3 pcDNA3.1-T2A-TagRFP (NM_001035240.2/α3) | YouBio | N/A |
| mCpne6$_{WT}$ pcDNA3.1-3xFlag-C | YouBio | N/A |
| mCpne6$_{1-269}$ pcDNA3.1-3xFlag-C | YouBio | N/A |
| mCpne6$_{134-557}$ pcDNA3.1-3xFlag-C | YouBio | N/A |
| mCpne6$_{269-557}$ pcDNA3.1-3xFlag-C | YouBio | N/A |
| mCpne6$_{134-269}$ pcDNA3.1-3xFlag-C | YouBio | N/A |
| mCpne6$_{1-151}$ pcDNA3.1-3xFlag-C | YouBio | N/A |
| mTrpa1 pcDNA3.1-3xFlag-C | From Chuan Wang lab | N/A |
| mTrpv1 pcDNA3.1-TagGFP-C | From Jinlong Qi lab | N/A |
| mTrpm2 pcDNA3.1-3xFlag-C | From Linhua Zhang lab | N/A |

| Reagent/resource | Reference or source | Identifier or catalog number |
|---|---|---|
| **Antibodies** | | |
| Mouse anti-NF200 (IF 1:300) | GeneTex | Cat# GTX634289 RRID: AB_2888431 |
| Chicken anti-Peripherin (IF 1:300) | Abcam | Cat# ab39374 |
| Guinea Pig IgG anti-TRPV1 (IF 1:300) | Neuromics | Cat# GP14100 RRID: AB_1624142 |
| Rabbit anti-TRPA1 (IF 1:300) | Alomone Labs | Cat# ACC-037 |
| Rabbit anti-TRPM3 (IF 1:500) | Alomone Labs | Cat# ACC-050 |
| Rabbit anti-TRPM3 (WB 1:1000) | Bioss | Cat# bs-9046R |
| Mouse anti-Copine-6 (IF 1:200) | Santa Cruz | Cat# sc-136357 |
| Rabbit anti-Copine-6 (IF 1:300; WB 1:1500) | Proteintech | Cat# 13782-1-AP RRID: AB_2292172 |
| Mouse anti-DDDDK-tag (IP 2 µg; IF 1:1000; WB 1:8000) | MBL | Cat# M185-3L |
| Rabbit anti-DYKDDDDK tag (IF 1:600) | Cell Signaling Technology | Cat# 14793T |
| Mouse anti-RFP (IP 2 µg; IF 1:300; WB 1:1000) | GeneTex | Cat# GTX628545 |
| Goat anti-rabbit IgG (WB 1:8000) | Cell Signaling Technology | Cat# 5151 |
| Goat anti-mouse IgG (WB 1:8000) | Cell Signaling Technology | Cat# 5257 |
| Goat anti-Guinea Pig IgG (IF 1:500) | Invitrogen | Cat# A-21450 RRID: AB_2535867 |
| Goat anti-Chicken IgY (IF 1:500) | Invitrogen | Cat# A-32932 RRID: AB_2762844 |
| Donkey anti-Mouse IgG (IF 1:500) | Invitrogen | Cat# A-21202 RRID: AB_141607 |
| Goat anti-Rabbit IgG (IF 1:500) | Invitrogen | Cat# A-11008 RRID: AB_143165 |
| Donkey anti-Rabbit IgG (IF 1:500) | Invitrogen | Cat# A-11008 RRID: AB_2534017 |
| **Oligonucleotides and other sequence-based reagents** | | |
| rat-Cpne6-F (5'-3') | GCCTGCCCATGTCCATCATCATC | N/A |
| rat-Cpne6-R (5'-3') | ACCATCGTCTCCGTCCAGTAGC | N/A |
| rat-Cpne1-F (5'-3') | GACGCTGACTCTACCCTTGATGTTG | N/A |
| rat-Cpne1-R (5'-3') | CACGGCTGTCCTTTAACTCCTGAG | N/A |
| rat-Cpne2-F (5'-3') | CTTTCTGCTCAGGCGTGGATGG | N/A |
| rat-Cpne2-R (5'-3') | GTTGCTGTCTGCTGCTCTGTGG | N/A |
| rat-Cpne3-F (5'-3') | GGTGTTATCGTTGTGAAGCATTGTG | N/A |
| rat-Cpne3-R (5'-3') | ACCGTTGGAGCCAGTGAAGTC | N/A |
| rat-Cpne4-F (5'-3') | GTGGACAGGACGGAAGTGATTCG | N/A |
| rat-Cpne4-R (5'-3') | GTCAGCCTCCTTCAGCCCATTATG | N/A |
| rat-Cpne5-F (5'-3') | ATCGTTCAGTTTGTGCCCTTCAGG | N/A |
| rat-Cpne5-R (5'-3') | GCCTTCATGTAGGACACCAGTTGG | N/A |
| rat-Cpne7-F (5'-3') | CCTCTGAAGTGCCTGGTCTGG | N/A |
| rat-Cpne7-R (5'-3') | TCTGCGAAGGTGGTGGTGAAG | N/A |
| rat-Cpne8-F (5'-3') | AGAATCCTTACTGTGACGGCATTG | N/A |
| rat-Cpne8-R (5'-3') | ATTACGGGAGCAAAGTTGGTTGG | N/A |
| rat-Cpne9-F (5'-3') | GTGTGGAACCATACTGCTGACTG | N/A |
| rat-Cpne9-R (5'-3') | GTGCCGTCTTCATTGCTCCTG | N/A |
| rat-Gapdh-F (5'-3') | TTGTGCAGTGCCAGCCTC | N/A |
| rat-Gapdh -R (5'-3') | TGAACTTGCCGTGGGTAGAG | N/A |
| mouse-Cpne1-F (5'-3') | AACAACCTGAACCCTACCTGGAAG | N/A |
| mouse-Cpne1-R (5'-3') | GAGCCGTCACTGTCATAGTCTGAG | N/A |

| Reagent/resource | Reference or source | Identifier or catalog number |
|---|---|---|
| mouse-Cpne2-F (5'-3') | CATCCCCTTGGGTTCTCAGTG | N/A |
| mouse-Cpne2-R (5'-3') | GTTGACGGCTGTTTCTGTTCT | N/A |
| mouse-Cpne3-F (5'-3') | TCCACAGTACCAGCCAGCATTC | N/A |
| mouse-Cpne3-R (5'-3') | GAGCCTAACCTCAGCAGGAAGC | N/A |
| mouse-Cpne4-F (5'-3') | CTTTATGGTCCCACCAACATCG | N/A |
| mouse-Cpne4-R (5'-3') | CTCCCTTGGGTGACCTTAGAA | N/A |
| mouse-Cpne5-F (5'-3') | GATCACGGTGTCATGCAGGAA | N/A |
| mouse-Cpne5-R (5'-3') | CTCTCGCCATTGCTTGTTCTC | N/A |
| mouse-Cpne6-F (5'-3') | GTGTGGTGAGTGATATGGCAGAGAC | N/A |
| mouse-Cpne6-R (5'-3') | TCCGTCCAGTAGCCTCATGTCAG | N/A |
| mouse-Cpne7-F (5'-3') | GCCTCTGAAGTGCCTGGTCTG | N/A |
| mouse-Cpne7-R (5'-3') | GCCTGCTGCTCCTCCTCAAAG | N/A |
| mouse-Cpne8-F (5'-3') | ACATTGGGGGAGATTGTTGGT | N/A |
| mouse-Cpne8-R (5'-3') | ACTTCTGTCTTGTGGCAAATTGT | N/A |
| mouse-Cpne9-F (5'-3') | AATTACTGTGTCGTGCCGGAA | N/A |
| mouse-Cpne9-R (5'-3') | TTGTCAATCACCTCGGTTCGT | N/A |
| mouse-Gapdh-F (5'-3') | AACATCAAATGGGGTGAGGCC | N/A |
| mouse-Gapdh-R (5'-3') | GTTGTCATGGATGACCTTGGC | N/A |
| mouse-Trpm3-F (5'-3') | ACATCTTCGGCGTGAACAAGTATC | N/A |
| mouse-Trpm3-R (5'-3') | TCATCAGCACCACCAGCATAATG | N/A |
| **Chemicals, enzymes and other reagents** | | |
| Rabbit IgG | Proteintech | Cat# B900610 |
| Mouse IgG | Solarbio | Cat# SP031 |
| Protein A/G PLUS-Agarose | Santa Cruz | Cat# sc-2003 |
| Capsaicin | MedChemExpress | Cat# HY-10448 |
| Complete Freund's adjuvant | Sigma-Aldrich | Cat# F5881 |
| Adenosine 5'-diphosphoribose sodium | MedChemExpress | Cat# HY-100973A |
| Allyl isothiocyanate | Alfa Aesar | Cat# L02901 |
| CIM0216 | MedChemExpress | Cat# HY-110220 |
| Pregnenolone monosulfate sodium | MedChemExpress | Cat# HY-110189 |
| Isosakuranetin | GLPBIO | Cat# GC30081 |
| Minute™ Plasma Membrane/Protein Isolation and Cell Fractionation Kit | Invent | Cat# SM-005 |
| **Software** | | |
| GraphPad Prism | https://www.graphpad.com | |
| ImageJ | https://imagej.net/ij/index.html | |
| IBM SPSS | https://www.ibm.com/spss | |
| Adobe Illustrator | https://www.adobe.com/products/illustrator.html | |
| **Other** | | |

## Animals

Animal experiments were performed using adult C57BL/6J (Beijing Vital River Laboratory Animal Technology Co. Ltd.) mice aged 6–12 weeks. Adult Sprague-Dawley (Beijing Vital River Laboratory Animal Technology Co. Ltd.) or Wistar (Leeds University colony) rats (150–250 g) were also used. Male animals were used, unless specified otherwise. Animals were given ad libitum access to food

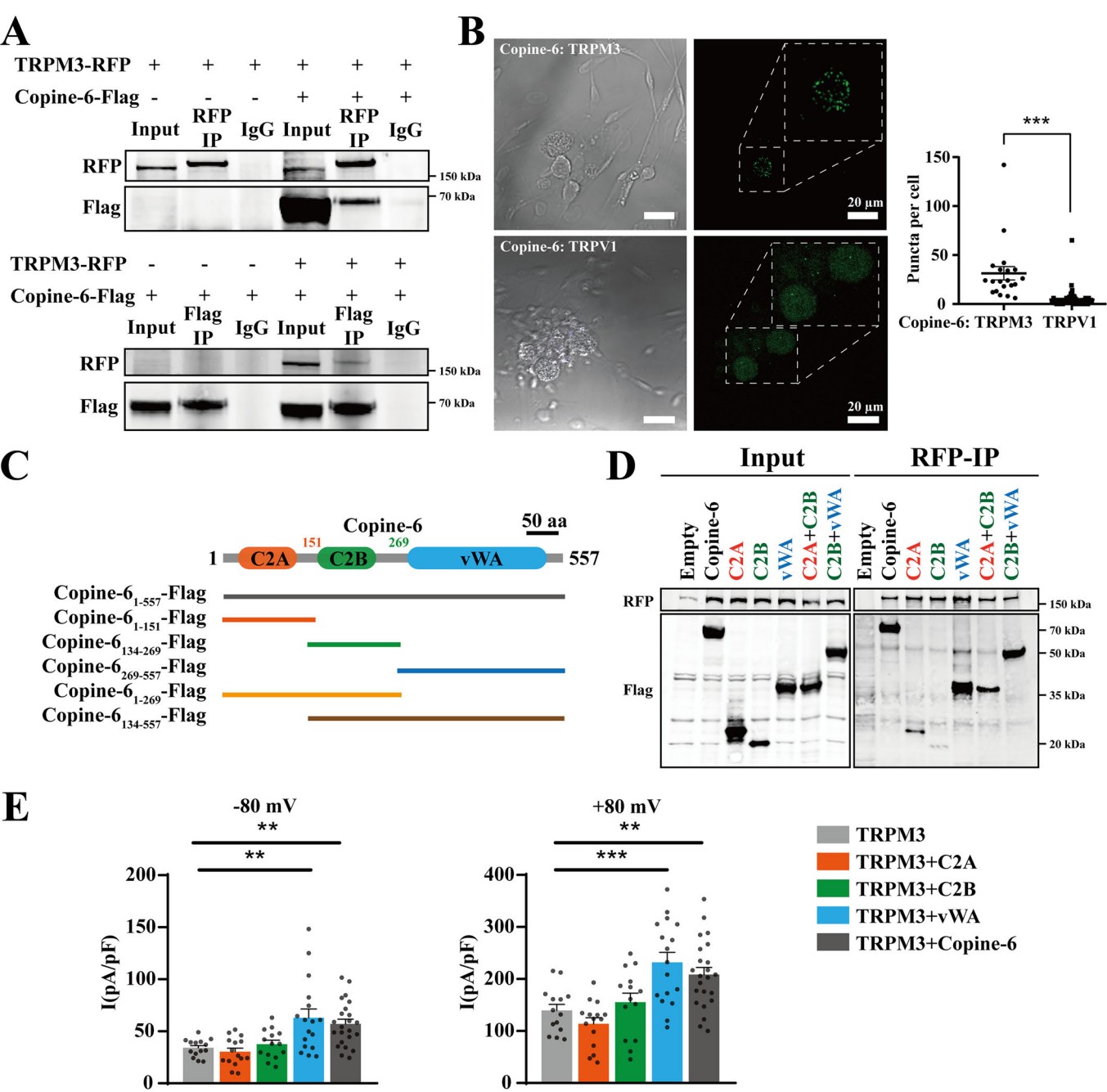

**Figure 9. Copine-6 physically interacts with TRPM3.**

(A) Example co-immunoprecipitation (Co-IP) of TRPM3-RFP and Copine-6-Flag in HEK293T cells. HEK293T cells transiently transfected with TRPM3-RFP alone, Copine-6-Flag alone, or both, TRPM3-RFP and Copine-6-Flag; immunoprecipitation from cell lysates was performed with anti-RFP or anti-Flag antibodies. (B) Proximity ligation assay (PLA) for Copine-6 and TRPM3 (top row) or Copine-6 and TRPV1 (bottom row) in rat DRG neurons. Fluorescent puncta produced by clustering of Copine-6 and TRPM3 or Copine-6 and TRPV1 are quantified in the scatter plot on the right. Copine-6/TRPM3: $n = 22$ positive neurons out of 60 observed (from three independent preparations); Copine-6/TRPV1: $n = 17$ positive neurons out of 60 observed (from three independent preparations); Data are shown as mean ± SEM; ***$P < 0.001$ (Mann–Whitney $U$ test). (C) Schematic diagram of Copine-6-Flag truncation constructs used in the next round of Co-IP experiments. (D) HEK293T cells were co-transfected with expression constructs encoding Flag-tagged Copine-6 truncation mutants together with TRPM3-RFP and the Co-IP performed. Left: western blot analysis of HEK293T cells lysates before immunoprecipitation. Right: analysis of anti-RFP immuno-precipitates using antibodies to Flag or RFP. (E) CHO cells were co-transfected with expression constructs encoding Flag-tagged Copine-6 deletion mutants (as shown in C), together with TRPM3. Currents were recorded with 5 µM CIM0216 using a voltage ramp from $-100$ mV to $+100$ mV. Comparison of peak current densities induced by CIM0216 in CHO cells at $-80$ and $+80$ mV. TRPM3, $n = 14$; TRPM3 + C2A, $n = 15$; TRPM3 + C2B, $n = 14$, TRPM3 + vWA, $n = 17$, TRPM3 + Copine-6: $n = 23$. Data are shown as mean ± SEM. $-80$ mV: TRPM3 vs. TRPM3 + vWA, **$P = 0.0014$; TRPM3 vs. TRPM3 + Copine-6, **$P = 0.009$. $+80$ mV: TRPM3 vs. TRPM3 + vWA, ***$P = 0.0004$; TRPM3 vs. TRPM3 + Copine-6, **$P = 0.0065$ (one-way ANOVA followed by Bonferroni multiple comparisons test). Source data are available online for this figure.

and water and housed in a controlled environment following the institutional guidelines and the Animal Research: Reporting of In Vivo Experiments (ARRIVE) guidelines (Percie du Sert et al, 2020). Animal experiments performed at Hebei Medical University were in accordance with the Animal Care and Ethical Committee of Hebei Medical University (Shijiazhuang, China; approval IACUC-Hebmu-2020007). Animal work carried out at the University of Leeds was approved by the University of Leeds Animal Welfare and Ethical Review Committee and performed under the UK Home Office License P40AD29D7 in accordance with the regulations of the UK Animals (Scientific Procedures) Act 1986.

$Cpne6^{+/-}$ mice on the C57BL/6 background were acquired from Cyagen Biosciences Inc. The design strategy was as follows. The Cpne6 gene (NCBI Reference Sequence: NM_001136057; Ensembl: ENSMUSG00000022212) is located on mouse chromosome 14 and consists of 17 exons. The ATG start codon is located on exon 2, while the TGA stop codon is located on exon 16 (Transcript: ENSMUST00000074225). To generate knockout ($Cpne6^{-/-}$) mice, exons 2 through 16 were targeted. Cas9 and guide RNA (gRNA) were co-injected into fertilized eggs to facilitate knockout mouse production. The resulting pups underwent genotyping using PCR, following specific primers:

KO-F (5'-CACTTCTATCCCAGTCCTCCAGTC-3');
KO-R (5'-GACACAGGTCACTGGAATCACAAG-3');
WT-F (5'-CATGTCAATCATCATCGTGGGTGT-3');
WT-R (5'-GACACAGGTCACTGGAATCACAAG-3').

The deletion spans 4997 bp and does not overlap with any other known genes.

The mice were born at Mendelian rates, exhibited normal fertility and were generally healthy. They also demonstrated normal motor capacity and coordination and showed no significant behavioral or morphological defects. Whenever possible, animals were randomly assigned to different experimental groups, and the investigators were blinded to animal treatment allocations.

## Cell cultures and transfection

DRG neurons were dissociated from male rats (80 g) or adult male mice (20–25 g). DRG from all spinal levels were removed and treated at 37 °C in Hank's Balanced Salt Solution (HBSS, Gibco) supplemented with collagenase (1 mg/mL, Worthington) and dispase (7.5 mg/mL, Sigma) for 30 min. Ganglia were then mechanically triturated, washed twice by centrifugation and resuspended by trituration in DMEM (Gibco) culturing media containing 10% fetal bovine serum (FBS, Gibco), 1% penicillin and streptomycin mixture (Gibco). This suspension was then plated on glass coverslips coated with 0.1 mg/mL poly-D-lysine (Gibco), and cultured at 37 °C in 5% $CO_2$ incubator. The neurons were used for electrophysiological recordings approximately 4 h to 24 h after plating.

HEK293T and CHO cells were freshly (less than 15 passages) purchased from Kunming Cell Bank (HE293T: Cat#KCB200744YJ; CHO: Cat#KCB83004YJ) and cultured at 37 °C in 5% $CO_2$ incubator using DMEM supplemented with 10% FBS, 1% penicillin and streptomycin mixture. For transfection, cells were cultured on 24-well plates or 10 cm-cell culture dishes and transfected with Lipo8000™ Transfection Reagent (Beyotime Biotechnology), according to the manufacturer's instructions. For patch-clamp recordings, 24 h after transfection, the cells were plated onto round glass slides; experiments were performed 48 to 96 h after transfection. As a marker for successfully transfected cells, cDNA-encoding GFP was co-transfected together with the other genes of interest. List of cDNA constructs used in this study is given in "Reagents and Tools "Table".

For gene silencing experiments, short hairpin RNA (shRNA) were used. The knockdown efficiency of four Cpne6-targeting shRNAs (Hippo bio) were tested in HEK293T cells overexpressing Cpne6; Lipo8000™ Transfection Reagent (Beyotime Biotechnology) was used according to instructions of manufacturer. The most effective sequence, shRNA2, was then tested on cultured DRG neurons as small interfering RNA (siRNA) with the same interference sequence; Lipofectamine RNAiMAX Transfection Reagent (Thermo Fisher Scientific) was used according to instructions of manufacturer. The shRNA2, as well as Scrambled shRNA were then used to produce viral constructs for in vivo Cpne6 knockdown (see below).

## Patch clamp recording

Whole-cell patch clamp recordings from mouse DRG neurons and transfected CHO cells were performed with a MultiClamp 700B amplifier/Digidata 1550 and pCLAMP 10.0 software (Molecular Devices) at room temperature. Cells were continuously perfused at approximately 2 mL/min with the extracellular bath solution. For recording TRPV1, TRPA1 and TRPM3 currents, extracellular solution contained (in mM): 140 NaCl, 3 KCl, 2 $CaCl_2$, 1.5 $MgCl_2$, 10 HEPES, and 10 glucose; the intracellular solution contained (in mM): 140 KCl, 1 $MgCl_2$, 5 EGTA, 10 HEPES. For recording TRPM2 currents, extracellular solution contained (in mM): 147 NaCl, 2 KCl, 2 $CaCl_2$, 1 $MgCl_2$, 10 HEPES, and 13 glucose (pH adjusted to 7.4 with NaOH); the intracellular solution contained (in mM): 147 NaCl, 1 $MgCl_2$, 0.05 EGTA, 10 HEPES (pH adjusted to 7.3 with NaOH). The recording electrodes (resistance 3–5 MΩ) were fabricated from borosilicate glass capillaries using a P-97 puller (Sutter) and fire-polished. Capacitance artifacts were canceled and series resistance compensated by 60–70%. Continuous gap-free voltage-clamp recordings were performed at a holding potential of -60 mV. TRPM2 and some TRPM3 currents were recorded by voltage-ramps from −150/−100 mV to +150/ + 100 mV delivered every 1/2 s from a holding potential of 0 mV.

## Quantitative PCR

Total RNA from cells/tissues was extracted using a commercial RNA isolation reagent (Trizol, Thermo Fisher). Isolated RNA was dissolved in 10–50 μL DEPC water and reverse-transcribed using an RT reagent kit (HiScript III RT SuperMix for qPCR (+ gDNA wiper), Vazyme, R323-01) and a thermal cycler (Mastercycler, Eppendorf). Quantitative PCR reaction was performed using a kit (ChamQ Universal SYBR qPCR Master Mix, Vazyme, Q711-02), and the fluorescent DNA was detected and quantified with an FQD-48A (A4) system (BIOER). List of primers used in this study is given in "Reagents and Tools Table".

## Western blot

Whole DRG tissue or HEK293T cells were washed three times with cold PBS (Gibco), and an appropriate volume of RIPA buffer (Thermo Fisher Scientific) containing 0.1 mM phenylmethanesulfonyl fluoride

(PMSF, GLPBIO) was then added. A tissue homogenizer was used at 70 Hz for 3 min to thoroughly disrupt the cells and the homogenates were incubated on ice for 30 min. Following this, centrifugation was performed at 14,000 × g for 30 min, and supernatants were transferred to 1.5-mL Eppendorf tubes to obtain the total protein lysate. Protein concentration was measured with BCA Protein Assay Kit (MULTI SCIENCES). The total protein lysates, cell membrane lysates (see below) or lysates that underwent co-immunoprecipitation (see below) were subjected to SDS-PAGE gels electrophoresis separation and transferred to 0.45-μm PVDF membranes (Millipore). The membranes were blocked by 5% non-fat milk (Servicebio) in TBST buffer (TBS buffer with 0.05% Tween 20) and incubated with the primary antibodies overnight at 4 °C. Next day, membranes were washed 3 times with TBST buffer and incubated with anti-rabbit IgG DyLight 800 conjugated secondary antibody (Cell Signaling Technology) or anti-mouse IgG DyLight 800 conjugated secondary antibody (Cell Signaling Technology) at room temperature for 2 h, followed with washing (TBST, three times). Finally, the detection process was carried out utilizing the Odyssey 9120 infrared imaging system.

## Co-immunoprecipitation

HEK293T cells were prepared and transfected as described in the previous section. Following three washes with ice-cold PBS (Gibco), the cells were lysed using cell lysis buffer for western blot and immunoprecipitation (Beyotime) containing 0.1 mM PMSF (GLPBIO) for 30 min on ice. The insoluble fraction was removed by high speed centrifugation at 14,000 × g for 30 min at 4 °C and the supernatant lysates were transferred to Eppendorf tubes. Protein concentration was measured with BCA Protein Assay Kit (MULTI SCIENCES). The total protein from each sample was precleaned and divided into three groups: 'input', 'immunoglobulin G (IgG)' and 'Co-IP'. The "input" groups were mixed with 5x loading buffer at a ratio of 4:1, heated at 95 °C for 5 min, and stored at −20 °C. The "Co-IP" groups were incubated with anti-Flag antibody (MBL, M185-3L, 2 μg) or anti-RFP antibody (GeneTex, GTX628545, 2 μg) at 4 °C for 4 h with rotation. The "IgG" groups were incubated with mouse IgG (Solarbio) or rabbit IgG (Proteintech) under the same conditions. Protein A/G PLUS-Agarose beads (Santa Cruz, 50 μL) were added to the "Co-IP" and "IgG" groups and incubated at 4 °C overnight with rotation. These beads were washed five times with PBS and centrifuged at 1000 × g at 4 °C for 5 min. The pellets were resuspended with 60 μL of 1× loading buffer, boiled for 5 min, and then subjected to western blot analysis.

## Immunofluorescence

DRG were either cryo-sectioned or embedded into gelatine and sectioned using a vibratome. For cryo-sectioning the procedure was as follows: lumbar DRG were isolated and fixed in 4% parafor-maldehyde (PFA, Biosharp) for 6 h before incubation at 4 °C overnight in a 30% sucrose solution. Tissues were then embedded in O.C.T. Compound (OCT, Sakura Finetek) and cryo-sectioned at a thickness of 10 μm. Sections were washed once with PBS, permeabilized for 60 min with 0.3% Triton X-100 and 3% bovine serum albumin (BSA) in PBS, then blocked for 30 min with 10% goat and/or donkey serum at 37 °C. The sections were incubated

overnight at 4 °C with primary antibody diluted in 0.1% Triton X-100 solution containing 1% BSA. The following day, sections were washed with PBS three times and incubated for 1.5 h at room temperature with the secondary antibodies. The sections were then washed three times with PBS and mounted with an enhanced antifade mounting medium.

For the vibratome sectioning the procedure was as follows. DRG were isolated and placed into 4% paraformaldehyde (PFA) for 1 h. Samples were then embedded in 10% gelatine and placed in 4% PFA for a further 4 h. Sections (40 μm thickness) were cut on a vibratome (Leica VT1000S) and placed in PBS. An antigen retrieval protocol was performed by incubating slices in Tris-EDTA buffer (10 mM Tris Base, 1 mM EDTA solution, 0.05% Tween 20, pH 9.0) for 30 min at 85 °C, before a further incubation of 30 min at room temperature. Sections were washed with PBS, permeabilized for 2 h in PBS containing 0.05% Tween 20, 0.25% Triton X-100 and 5% goat and/or donkey serum at room temperature. DRG sections were then incubated overnight at 4 °C with primary antibody diluted in PBS consisting of 50 mg/mL BSA. The following day, sections were washed with PBS three times and incubated for 2 h at room temperature with secondary antibodies in PBS with 10 mg/mL BSA. Three further washes with PBS were undertaken before mounting with mounting medium containing DAPI (Abcam).

## Fluorescence co-localization analysis

For quantification of Copine-6 distribution, the following approach was used. Regions of interest (ROIs) of DRG neuron cell bodies were detected based on the somatic immunofluorescence with the software Cellpose (Stringer et al, 2021). The masks obtained were loaded into Fiji and applied to images of immunofluorescence to acquire the number and area of neurons.

To detect membrane-localized immunofluorescence, the ROIs automatically detected by Cellpose were overlaid onto the Copine-6/TRPM3/TRPV1/TRPA1 fluorescence channels using the Fiji plug-in "Labels to ROIs" which allowed us to change the size of ROIs. Each original ROI (ROI-1) was shrunk to 80% of its initial scale to obtain the shrunken ROI (ROI-2) of a neuron (Fig. 8E). The "doughnut" ROI between the original ROI and the shrunken ROI (ROI-3) contains most of the plasma-membrane-associated immunofluorescence within the given plane. Mean fluorescence intensity was measured for each region using Fiji and relative changes in plasma-membrane-associated immunofluorescence were expressed as $F_{ROI3}/F_{ROI2}$.

## RNAscope

Rat DRG harvesting, fixing, and sectioning were performed similarly to that for immunofluorescence labeling. RNAscope was performed following the manufacturer's instructions using the RNAscope™ Intro Pack 2.5 HD Duplex Reagent Kit (Bio-Techne) in combination with RNAscope® Probe - Mm-Cpne6 (Cat #822931) and RNAscope® Probe - Mm-Trpm3 (Cat #459911). Brightfield images were acquired using a Zeiss Axioscan 7 or Invitrogen Evos M5000 imaging system; at least 3 sections from each animal were used for data analysis. Cells with more than 5 puncta or bright throughout colorization were classified as positive for specific mRNA expression.

## Membrane protein extraction

Cells/tissues were collected and plasma membrane protein was isolated with the Plasma Membrane Protein Isolation and Cell Fractionation kit (CAT# SM-005, Invent) according to the manufacturer's instructions. Briefly, the whole DRGs or HEK293T cells were washed three times with cold PBS, after which 500 μL of Buffer A was added, and the mixture was transferred to filter cartridges. A grinding rod was used to homogenize the tissue for 1 min, and homogenates were incubated on ice for 5 min with the lid open. The mixture was centrifuged at 16,000 × g for 30 s. and filter cartridges discarded. The pellets were vortexed vigorously for 10 s, followed by centrifugation at 700 × g for 1 min. The supernatants were transferred to a new 1.5 mL Eppendorf tubes and centrifuged at 16,000 × g for 30 min at 4 °C. The supernatants were discarded, and pellets resuspended in 200 μL of Buffer B, mixed by vortexing and centrifuged at 7800 × g for 5 min at 4 °C. The supernatants were then transferred to a new 2.0 mL Eppendorf tubes and 1.6 mL of ice-cold PBS was added and mixed gently. Finally, samples were centrifuged at 16,000 × g for 30 min, resulting in a pellet that represented the plasma membrane-enriched fraction, which was dissolved using Minute™ Denaturing Protein Solubilization Reagent WA-009 (Invent). Protein concentration was measured with BCA Protein Assay Kit (MULTI SCIENCES). Antibody against $Na^+$-$K^+$-ATPase was used as loading control.

## Viral injections

AAV9-U6-Copine-6 shRNA2-EGFP, AAV9-U6-Scramble shRNA-EGFP, AAV9-hSyn-mCopine-6-EGFP and AAV9-hSyn-EGFP virions were obtained from Brain Case Co. Ltd. (Shenzhen, China). AAV virions were injected into the right-side L4 and L5 DRG of SD male rats or L3 and L4 DRG of C57BL/6 male mice. Under deep anesthesia (rat, 1.25% tribromoethanol, 10 mL/kg, i.p.; mouse, 1.25% tribromoethanol, 0.2 mL/10 g, i.p.). DRGs were exposed by removal of both spinous and transverse processes of the vertebra bone. The microinjector (Hamilton) was inserted into the ganglion to a depth of approximately 300 μm from the exposed surface. The virion solution ($1 \times 10^{13}$ viral genomes/mL; 2 μL) was injected at a rate of 0.25 μL/min, and the needle was removed 5 min after the injection was complete. The muscles overlying the spinal cord were loosely sutured together, and the wound was closed. Animals were used for further experiments 4 weeks after injection.

## Complete Freund's adjuvant (CFA) inflammatory pain model

To induce chronic inflammatory pain, mice were injected with 25 μL of complete Freund's adjuvant (Sigma) in the glabrous surface of the right hind paw, and sterile 0.9% NaCl was injected into the control mice. Same procedure was used for rats but 50 μL of CFA/saline was injected.

## Spared nerve injury (SNI) model

SNI surgery was performed as follows. The animals were sterilized with alcohol and operated under deep isoflurane anesthesia. The hair of the right hind leg was clipped, and the skin was cut at the sciatic tubercle. The muscle layer was separated by blunt dissection until exposure of the sciatic nerve and three distal branches. The peroneal and tibial branches were tightly ligated with 5–0 silk and transected below the ligature, and a 2–3 mm section distal to the ligature was removed, leaving the sural nerve intact. The muscle tissue and the skin were closed with sutures.

## Behavioral tests

In all tests, animals were habituated to the testing environment for at least 30 min before behavioral testing.

### von Frey test

The mechanical threshold was measured using calibrated von Frey filaments on a metal mesh platform. Von Frey filaments were applied to the right hind paw of the animals. The criterion for positive reaction was a sudden lift or lick of the tested foot. The lowest filament force that elicited positive reaction more than three times out of five times tests was defined as the mechanical threshold.

### Hargreaves test

Thermal withdrawal latency was tested by a radiant heat source (PL-200, Taimeng Co.). The intensity of the radiant heat source was maintained at $20 \pm 0.1\%$ for rats and $15 \pm 0.1\%$ for mice. Animals were placed individually into Plexiglas cubicles placed on a transparent glass surface. The light beam from heat lamp, located below the glass, was directed at the plantar surface of hind paw. The withdrawal latency was recorded when the right hind paw withdrew from the thermal radiometer. Animals were tested three times and the final results was the average of the three trials. The cutoff time was 30 s to prevent scalding.

### Hot plate test

In the hot plate test, animals were placed on a hot plate apparatus (UGO Basile 35150 Hot/Cold Plate) set at a specific temperature of 45, 50, 52 or 55 °C, and the latency to the first jump or lifting/licking the paw was recorded.

### Randall–Selitto test

Rats were gently restrained in the hand of the experimenter and a pinching force was applied to the hind paw using a Randall–Selitto device (SMALGO, RWD). A response was scored by any visible flinching of the hindlimb and the value of the force recorded.

### Dry ice test

Dry ice was chipped into small pieces and tightly packed into a 30 mL disposable syringe. The front end of the syringe was trimmed by 1 cm, and the dry ice cylinder was pushed out by 1 cm. Prior to testing, the animals were placed in a plastic box with the transparent glass bottom for at least 15 min. During the test, the dry ice cylinder was gently pressed against the glass plate beneath the hind paw of each animal, and the time from the onset of dry ice stimulation to the withdrawal of the hind paw was recorded. Each animal underwent three trials, spaced 5 min apart, and the average withdrawal time from these trials was calculated.

 

## Rotarod test

The rotarod apparatus was set to rotation speed of 40 r/min. In the three days preceding the test, the mice underwent a daily 10 min adaptability training session on the rotating rod. For the test, mice were placed at the onset of rotating rod with the head facing the direction of rest of the rod. The time taken for each animal to fall off the rod was recorded.

## Balance beam test

The rats were carefully placed on a stainless steel balance beam, measuring 2 cm in width and 1.5 m in length, and their balance ability was assessed over a 2-min period. The scoring criteria were as follows: (1) successfully crossing the balance beam or standing firmly without shaking for the entire 2 min; (2) shaking side to side while crossing the balance beam or shaking side to side on the balance beam for 2 min without falling; (3) sliding to one side but staying on the beam for the entire 2 min; (4) standing on the beam for less than 2 min before falling off; (5) attempting to stand on the beam but falling within seconds.

## Ink footprint

Non-toxic black ink was applied to the animal's hind paws, and the animal was placed on a track with blotting paper at the bottom. Gait prints were recorded, and the lateral swing distance and stride length were measured.

## Nocifensive behavior test

Mice were allowed to acclimatize for at least 20 min in a transparent observation chamber before the experiment. The right hind paw of the mice received an intraplantar injection of capsaicin (10 µg/25 µL, MedChemExpress), CIM0216 (10 nmol/25 µL, MedChemExpress) or vehicle (for capsaicin: 1.3% anhydrous ethanol in 25 µL saline; for CIM0216: 10% DMSO in 25 µL saline), and the spontaneous nocifensive responses (number of licking and lifting) were recorded using a video camera during 5 min. The videos were analyzed by an observer unaware of treatment allocations.

## Acute itch test

Mice were lightly anesthetized with isoflurane, and the dorsal neck area was shaved and disinfected. Compound 48/80 (100 µg/50 µL, dissolved in saline) or Chloroquine (CQ, 200 µg/50 µL, dissolved in saline) were injected intradermally into the dorsal neck area of the mice. Following the injection, the mice were placed in a recording system, and video-recorded for 30 min; the number of individual scratches was recorded.

## Chronic itch test

Mice were lightly anesthetized with isoflurane, and the dorsal neck area was shaved and disinfected. The back of the mouse neck was painted with 0.2 mL diphenylcyclopropenone (DCP, 1% dissolved in acetone). Six days after the initial sensitization, 0.2 mL 0.5% DCP was painted on the neck skin daily for 7 days. Scratching behavior was recorded by video for 30 min after daily application of DCP for the last 7 days.

## Fluorescence Ca$^{2+}$ imaging

Dissociated DRG neurons were cultured on 10 mm round glass coverslips, as described above. On the day of imaging, cells were incubated at room temperature in the dark for 45 min with Fluo-4-AM (2 µM, Thermo Fisher Scientific) and pluronic F-127 (0.02%, Sigma-Aldrich) in extracellular bath solution of the following composition (in mM): 145 NaCl, 5 KCl, 2 MgCl$_2$, 2 CaCl$_2$, 10 HEPES, 10 glucose (pH 7.4). The loading solution was then removed and cells were washed with extracellular solution. Coverslips were placed in a perfusion chamber, and normal extracellular solution was perfused at an approximate rate of 2 mL/min. Fluorescence imaging was performed on an inverted Olympus FV1200MPE fluorescence microscope. Regions of interest (ROIs) were used to select neurons from snapshots taken to record from and for further post hoc analysis. Recordings were analyzed in Microsoft Excel and Prism software.

## Proximity ligation assay (PLA)

PLA was performed as described previously (Jin et al, 2013; Shah et al, 2020). Briefly, proteins of interest within a cell were labelled with specific primary antibodies and then treated with PLA probes (Duolink, Sigma-Aldrich), which are secondary antibodies conjugated with short DNA oligos. If two proteins reside within less than 30-40 nm of each other, connector oligonucleotides facilitate the formation of a single-stranded DNA circle between the two secondary probes, a unique new DNA sequence is amplified, and a colour reaction is developed (Weibrecht et al, 2010). DRG cultures were plated onto 10 mm glass coverslips in six well plates and left to culture for 24–48 h. Cells were fixed with 4% ice-cold PFA for 20 min and permeabilized with buffer composed of 0.05% Tween 20 and 0.25% Triton X in PBS for 1 h at room temperature. A hydrophobic barrier pen was used to delimit the reaction area by drawing around the coverslip to stop the 'meniscus' droplet from breaking during incubations. Cells were then blocked for 30 min at 37 °C (using Duolink blocking buffer) and incubated with primary antibodies for 12–16 h at 4 °C. Following washing, the oligo-conjugated secondary antibodies (anti-mouse MINUS and anti-rabbit PLUS PLA probes, Sigma-Aldrich) were added for 1 h at 37 °C. Ligation and amplification was performed according to the manufacturer's instructions. Finally, the samples were washed and mounted with Duolink mounting medium with DAPI (Sigma-Aldrich). Imaging was performed using a Carl Zeiss LSM880 inverted confocal microscope and images processed by Zeiss ZEN software. Cells containing ≥5 puncta were considered positive.

## Statistical analysis

Sample size estimations were made using NC3Rs Experimental Design Assistant (https://eda.nc3rs.org.uk/) based on effect sizes and variability observed in previous experiments. Animals were allocated to the groups randomly and codified, the operators were unaware of the code. All data are given as mean ± SEM. Paired or unpaired *t* test was used to compare the means of two groups when

the data were normally distributed, if data failed normality test, Mann–Whitney test was used. Multiple groups were compared using one- or two-way ANOVA or repeated-measures ANOVA, depending on experimental setup; Bonferroni post hoc test was used for comparison between groups. For data failing normality test Kruskal–Wallis ANOVA with Mann–Whitney post hoc test was used. Proportions were tested for significance using Fisher's exact test. Statistical analyses were performed using GraphPad Prism or IBM SPSS Statistics.

## Data availability

This study includes no data deposited in external repositories. Source data and metadata used for figure preparation are included as supplement.

The source data of this paper are collected in the following database record: biostudies:S-SCDT-10_1038-S44318-025-00487-0.

## Peer review information

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

## Acknowledgements

We thank Stephen Milne for expert technical assistance, Jasper Morley and Simona Kent-Saisch for help with bioinformatics analysis of Copine-6 expression, we thank Isabella Moffitt, Louis Orton, Tobian Catsburg and Osato Eghi-Guobadia for help with immunohistochemistry. This work was supported by the by the Wellcome Trust Investigator Award 212302/Z/18/Z and Medical Research Council Project Grant MR/V012738/1 to NG. This project has received funding from the European Union's Horizon 2020 research and innovation programme under the Marie Skłodowska-Curie Actions grant agreement No 956477 (PIANO). Support was provided by the Hebei Province Talent and Intelligence Introduction Project Award to HG and NG; Central Guiding Local Science and Technology Development Fund Project (236Z7723G) to HG, Hebei Natural Science Foundation award (H2022206515) to HG, National Natural Science Foundation of China grant (81871027) to HG and NG; Key laboratory of Neural and Vascular Biology, Ministry of Education of China project NV20230001 to HG.

## Author contributions

**Yiting Gao**: Conceptualization; Investigation; Visualization; Writing—original draft; Writing—review and editing. **Shengxiang Yan**: Investigation; Visualization; Methodology; Writing—review and editing. **Zhongyang Zhang**: Investigation. **Jieyao Zhang**: Investigation. **Meng Yang**: Investigation. **Shihab Shah**: Investigation; Visualization; Methodology. **Sofia Figoli**: Investigation; Visualization; Methodology. **Qi Jing**: Investigation. **Haixia Gao**: Conceptualization; Supervision; Funding acquisition; Validation; Project administration; Writing—review and editing. **Nikita Gamper**: Conceptualization; Supervision; Funding acquisition; Validation; Visualization; Writing—original draft; Project administration; Writing—review and editing.

Source data underlying figure panels in this paper may have individual authorship assigned. Where available, figure panel/source data authorship is listed in the following database record: biostudies:S-SCDT-10_1038-S44318-025-00487-0.

## Disclosure and competing interests statement

The authors declare no competing interests.

# Expanded View Figures

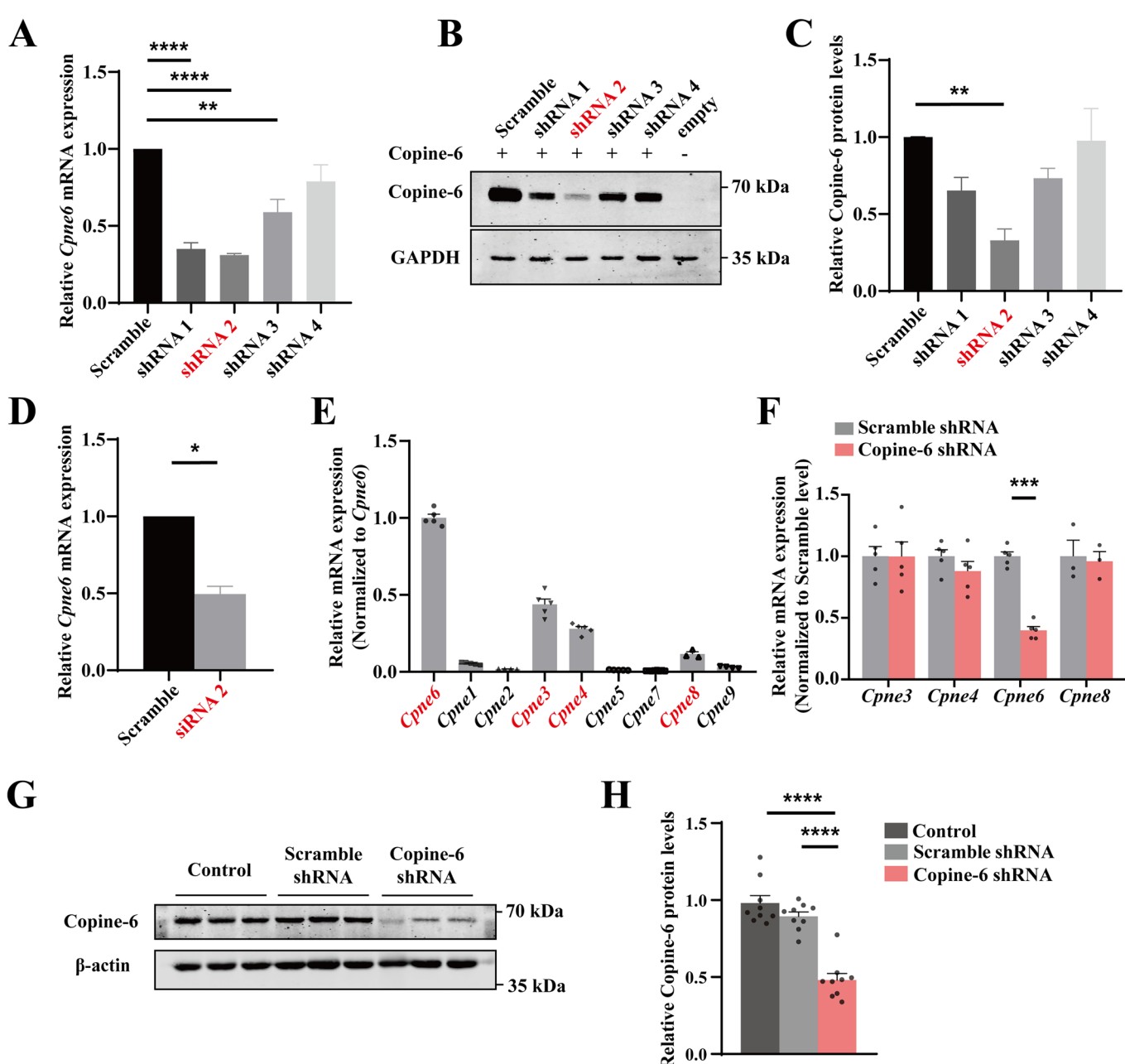

**Figure EV1. Validation of *Cpne6* knockdown efficiency in vitro and in vivo in the rat.**

(A) Summary data of *Cpne6* mRNA knockdown by four *Cpne6*-targeting shRNAs in HEK293T cells overexpressing *Cpne6*; $n = 4$; Data are shown as mean ± SEM; Scramble vs. shRNA1, ****$P < 0.0001$; Scramble vs. shRNA2, ****$P < 0.0001$; Scramble vs. shRNA3, **$P = 0.0014$ (one-way ANOVA followed by Bonferroni multiple comparisons test). (B) Example of western blot assay for Copine-6 expression in HEK293T cells co-transfected with Copine-6 and four shRNA constructs used. (C) Quantification of the Copine-6 protein expression knockdown by Copine-6 shRNAs in HEK293T cells (from the experiments as these shown in (B); $n = 4$; Data are shown as mean ± SEM; **$P = 0.0024$ (one-way ANOVA followed by Bonferroni multiple comparisons test). (D) Knockdown of *Cpne6* mRNA by siRNA2 in cultured DRG neurons; $n = 4$; Data are shown as mean ± SEM; *$P = 0.029$ (Mann–Whitney $U$ test). (E) The mRNA expression levels of *Cpne1-9* genes in the whole DRG of rats, relative to *Cpne6*. Genes displaying considerable expression are highlighted in red. *Cpne1, 3-6*, $n = 5$; *Cpne2, 9*, $n = 4$; *Cpne7*, $n = 8$; *Cpne8*, $n = 3$; Data are shown as mean ± SEM. (F) qPCR analysis of the *Cpne3, 4, 6* and *8* mRNA in the whole DRG of rats injected with AAV9-U6-Copine-6-shRNA-EGFP or AAV9-U6-Scramble-shRNA-EGFP (45 days after viral injection into the DRG); *Cpne3, 4, 6*, $n = 5$; *Cpne8*, $n = 3$. Data are shown as mean ± SEM. ***$P < 0.001$ (two-tailed independent $t$ test). (G) Example of western blot assay for Copine-6 expression in the DRGs of rats with viral DRG-targeted shRNA knockdown of *Cpne6*. (H) Quantification of the Copine-6 protein expression in the DRG (from the experiments as these shown in G); $n = 9$; Data are shown as mean ± SEM; ****$P < 0.0001$ (one-way ANOVA followed by Bonferroni multiple comparisons test). Source data are available online for this figure.

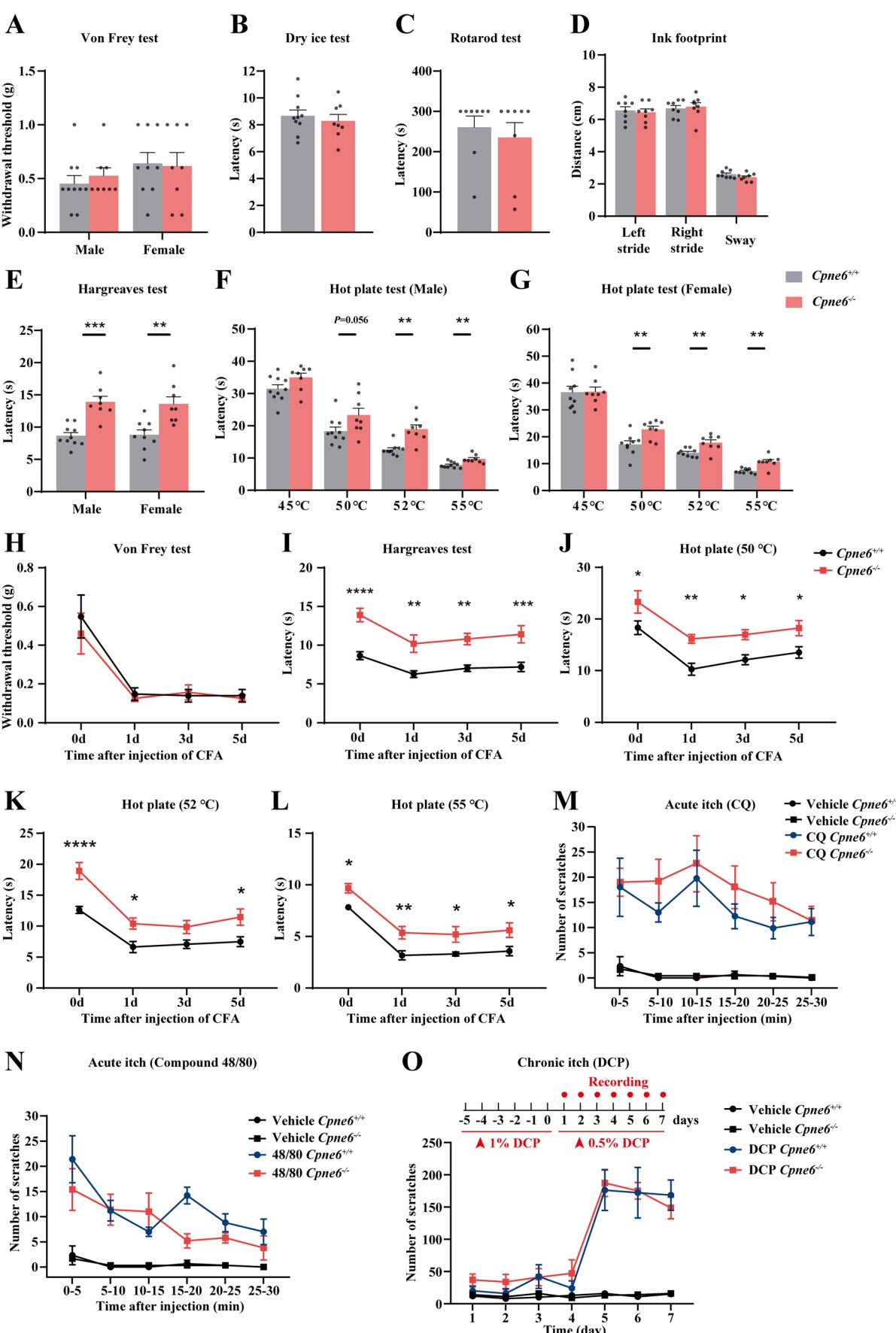

◄ **Figure EV2.  Additional behavioral characterization of *Cpne6*$^{-/-}$ mice.**

(A–F) The somatosensory and motor coordination tests were performed on *Cpne6*$^{+/+}$ and *Cpne6*$^{-/-}$ mice, as follows: Von Frey test (A), dry ice test (B), rotarod test (C), ink footprint test (D), Hargreaves test (E), hot plate test (F, G). (A, E–G): $n = 10$ mice in male *Cpne6*$^{+/+}$ group, $n = 9$ mice in female *Cpne6*$^{+/+}$ group and $n = 8$ mice in *Cpne6*$^{-/-}$ groups of either gender. (B) $n = 10$ mice in in *Cpne6*$^{+/+}$, $n = 8$ mice in *Cpne6*$^{-/-}$ group. (C, D) $n = 8$ mice in both groups. Data are shown as mean ± SEM. (E) Male, ***$P < 0.001$; Female, **$P = 0.003$. (F) 50 °C, $P = 0.056$; 52 °C, **$P = 0.001$; 55 °C, **$P = 0.002$. (G) 50 °C, **$P = 0.007$; 52 °C, **$P = 0.005$; 55 °C, **$P = 0.001$ (two-tailed independent *t* test). (H–L) Knockout of *Cpne6* reduced CFA-induced heat hyperalgesia (I–L) but not mechanical allodynia (H); $n = 10$ mice in all *Cpne6*$^{+/+}$ groups and $n = 8$ mice in all *Cpne6*$^{-/-}$ groups. Data are shown as mean ± SEM. I, 0 d, ****$P < 0.0001$; 1 d, **$P = 0.0012$; 3 d, **$P = 0.0019$; 5 d, ***$P = 0.005$. J, 0 d, *$P = 0.0308$; 1 d, **$P = 0.0078$; 3 d, *$P = 0.038$; 5 d, *$P = 0.0465$. (K) 0 d, ****$P < 0.0001$; 1 d, *$P = 0.025$; 5 d, *$P = 0.0168$. (L) 0 d, *$P = 0.0406$; 1 d, **$P = 0.0099$; 3 d, *$P = 0.0334$; 5 d, *$P = 0.0198$. Two-way ANOVA with Bonferroni multiple comparisons test. (M–O) Acute and chronic itch tests. (M) Histamine-independent acute itch model. Chloroquine (CQ) (200 µg/50 µL, vehicle: Saline) was intradermally injected into dorsal neck area and number of scratching attempts quantified during 30 min period. $n = 3$ mice in each vehicle group and $n = 8$ mice in each CQ group. Data are shown as mean ± SEM. (N) Histamine-dependent acute itch model. Compound 48/80 (100 µg/50 µL, vehicle: Saline) was intradermally injected into dorsal neck area and number of scratching attempts quantified during 30 min period. $n = 3$ mice in each vehicle group and $n = 5$ mice in each 48/80 group. Data are shown as mean ± SEM. (O) Chronic itch model. Shown on the top is the experimental timeline. Diphenylcyclopropenone (DCP, 1%, then 0.5% in acetone, as indicated) was applied to the dorsal neck area (see "Methods") and scratching behavior recorded for 30 min after each application during seven consecutive days; vehicle: acetone; $n = 3$ mice in each vehicle group and $n = 5$ mice in each DCP group. Data are shown as mean ± SEM. Source data are available online for this figure.

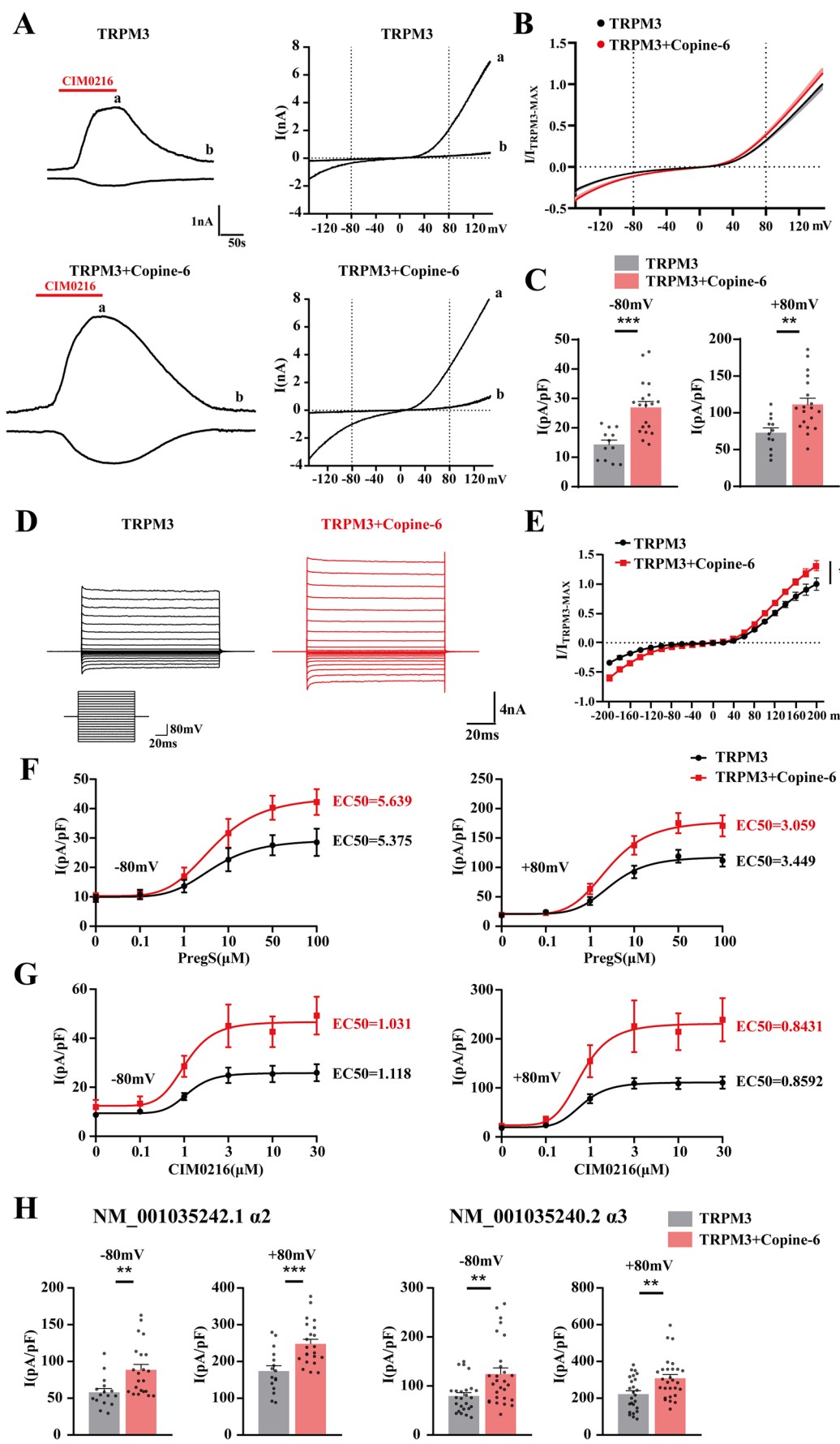

**Figure EV3. Expression of Copine-6 increases TRPM3-mediated currents.**

(A) Time course (left) and I-V relationship (right) of a whole-cell currents induced by CIM0216 (5 µM) in CHO cells expressing TRPM3 (top) or TRPM3 + Copine-6 (bottom). (B) Voltage dependence of CIM0216-induced currents in TRPM3-expressing CHO cells with (red) and without (black) Copine-6 co-expression. Currents were induced by voltage ramps from −150 to +150 mV in 900 ms. Current-voltage relationships normalized to $I_{TRPM3}$ at +150 mV are shown. TRPM3, $n = 12$; TRPM3 + Copine-6, $n = 19$. (C) Comparison of 5 µM CIM0216-induced current densities in CHO cells expressing TRPM3 (NM_177341.4/γ3), either alone or together with Copine-6 at −80 or +80 mV. TRPM3, $n = 12$; TRPM3 + Copine-6, $n = 19$; Data are shown as mean ± SEM; −80 mV, ***$P < 0.001$; +80 mV, **$P = 0.003$ (two-tailed independent t test). (D) Current traces obtained with a voltage step protocol (inset on the bottom-left) in CHO cells transfected with TRPM3 alone (black) or co-transfected with Copine-6 (red) after stimulation by 5 µM CIM0216. (E) Voltage dependence of CIM0216-induced currents in TRPM3-expressing CHO cells with (red) and without (black) Copine-6 co-expression. Current-voltage relationships normalized to $I_{TRPM3}$ at +200 mV are shown. TRPM3, $n = 13$; TRPM3 + Copine-6, $n = 19$; Data are shown as mean ± SEM; *$P = 0.0188$ (two-way ANOVA followed by Bonferroni multiple comparisons test). (F) Concentration-response curves for Pregnenolone monosulfate sodium (PregS) in CHO cells transfected with TRPM3 alone (black) or co-transfected with Copine-6 (red) at −80 mV (left) and +80 mV (right); recordings as these shown in (A). The $EC_{50}$ values are shown next to the curves. Current densities (pA/pF) represent the peak values of the PregS-induced currents. The number of recordings, n (from low to high concentration of PregS, respectively): 22, 18, 17, 20 22, and 20 for cells expressing TRPM3 alone (black); 20, 20, 20, 20, 19 and 18 for cells expressing TRPM3 and Copine-6 (red). Data are shown as mean ± SEM. All curves were fitted with Four-Parameter Logistic function. (G) Similar to (F), but CIM0216 concentration-response curves are plotted. The number of recordings, n (from low to high concentration of CIM0216, respectively): 17, 16, 16, 17 15, and 12 for cells expressing TRPM3 alone (black); 17, 17, 17, 14, 13 and 13 for cells expressing TRPM3 and Copine-6 (red). Data are shown as mean ± SEM. (H) Comparison of 5 µM CIM0216-induced current densities in CHO cells expressing TRPM3 splice variants (NM_001035242.1/α2 or NM_001035240.2/α3), either alone or together with Copine-6 at −80 or +80 mV. NM_001035242.1/α2: TRPM3, $n = 16$; TRPM3 + Copine-6, $n = 22$; NM_001035240.2/α3: TRPM3, $n = 25$; TRPM3 + Copine-6, $n = 28$. Data are shown as mean ± SEM. NM_001035242.1/α2: -80 mV, **$P = 0.002$; +80 mV, ***$P < 0.0001$. NM_001035240.2/α3: −80 mV, **$P = 0.003$; +80 mV, **$P = 0.0003$ (two-tailed independent Student's t-test). Source data are available online for this figure.

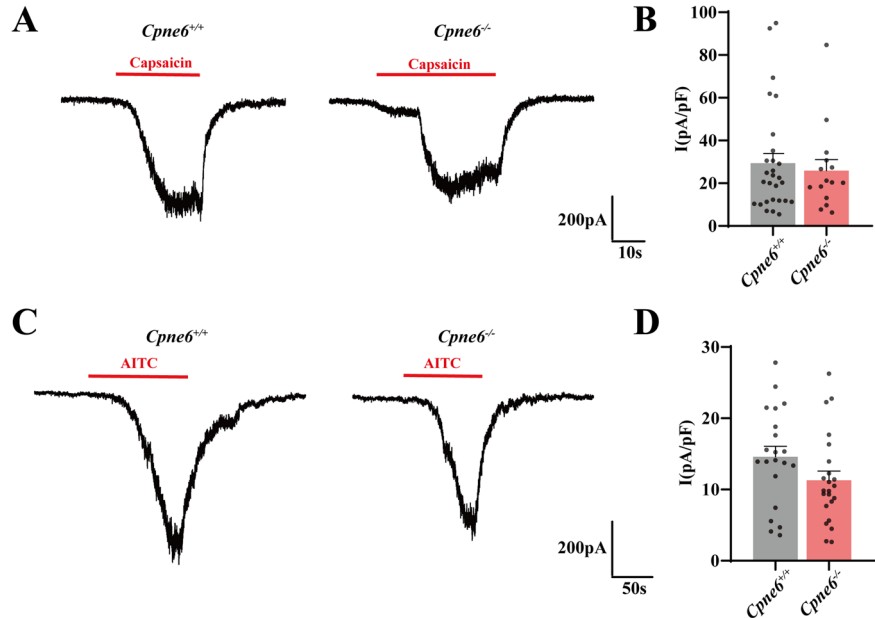

**Figure EV4. TRPV1 and TRPA1 currents in DRG neurons of *Cpne6*$^{+/+}$ and *Cpne6*$^{-/-}$ mice.**

(A) Example current traces evoked by capsaicin (1 μM) in DRG neurons from *Cpne6*$^{+/+}$ (left) or *Cpne6*$^{-/-}$ (right) mice. Recordings were performed using a gap-free protocol at a holding potential of −60 mV. (B) Comparison of capsaicin-evoked current densities in DRG neurons from *Cpne6*$^{+/+}$ ($n = 28$) and *Cpne6*$^{-/-}$ ($n = 15$) mice. Data are shown as mean ± SEM. (C) Example currents evoked by AITC (100 μM) in DRG neurons from *Cpne6*$^{+/+}$ (left) or *Cpne6*$^{-/-}$ (right) mice. Recordings were performed as in (A). (D) Comparison of AITC-evoked current densities in DRG neurons from *Cpne6*$^{+/+}$ ($n = 21$) and *Cpne6*$^{-/-}$ ($n = 23$) groups. Data are shown as mean ± SEM. Source data are available online for this figure.

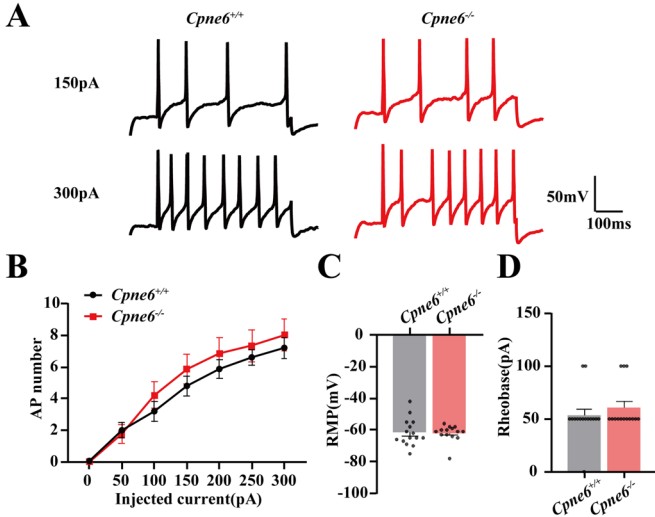

**Figure EV5.   Copine-6 deficiency has no impact on the general excitability of DRG neurons.**

(A) Representative current clamp recordings from dissociated DRG neurons from *Cpne6*⁺/⁺ (left) or *Cpne6*⁻/⁻ (right) mice. Firing is induced by 500 ms injections of either 150 (top) or 300 pA (bottom) depolarizing currents. (B) Summary of the numbers of action potentials elicited by depolarizing current steps from 0 to 300 pA (in 50 pA increments) elicited in DRG neurons from *Cpne6*⁺/⁺ ($n = 15$) or *Cpne6*⁻/⁻ ($n = 14$) mice; Data from 3 independent preparations. Recordings were made at room temperature. Data are shown as mean ± SEM. (C, D) Bar graphs and scatter plots showing the resting membrane potential (RMP; C) and rheobase (D) of DRG neurons from *Cpne6*⁺/⁺ ($n = 15$) or *Cpne6*⁻/⁻ ($n = 14$) mice; data from 3 independent preparations. Data are shown as mean ± SEM. Source data are available online for this figure.

