## [Peer Review File · The EMBO Journal]

Copine-6 is a TRPM3 escort protein controlling the sensitivity of sensory neurons to noxious heat

Yiting Gao, Shengxiang Yan, Zhongyang Zhang, Jieyao Zhang, Meng Yang, Shihab Shah, Sofia Figoli, Qi Jing, Haixia Gao, and Nikita Gamper

Corresponding author(s): Nikita Gamper (N.Gamper@leeds.ac.uk) , Haixia Gao (gaohx686@hebmu.edu.cn)

Review Timeline:

Submission Date:	7th Nov 24
Editorial Decision:	12th Dec 24
Revision Received:	23rd Apr 25
Editorial Decision:	15th May 25
Revision Received:	21st May 25
Accepted:	27th May 25

Editor: William Teale

Transaction Report:

Dear Prof. Gamper,

Thank you again for the submission of your manuscript entitled "Copine-6 is a TRPM3 escort protein controlling noxious heat sensitivity of sensory neurons" and for your patience during the review process. We have now received the reports from the referees, which I copy below.

As you can see from their comments, while referee #2 focusses mainly on technical issues, referee #1 would like to learn more about the functional specificity of the TRPM3-Copine-6 interaction. These comments will require your attention before your manuscript can be published in The EMBO Journal. Both referees, however, point out that your study is timely and potentially important.

Based on the overall interest expressed in the reports, I would like to invite you to address the comments of all referees in a revised version of the manuscript. I should add that it is The EMBO Journal policy to allow only a single major round of revision and that it is therefore important to resolve the main concerns at this stage. I believe the concerns of the referees are reasonable and addressable, but please contact me if you have any questions, need further input on the referee comments or if you anticipate any problems in addressing any of their points. I will be available from 2nd of January for a Zoom call to discuss these reports, should you wish to do so. Please, follow the instructions below when preparing your manuscript for resubmission.

I would also like to point out that as a matter of policy, competing manuscripts published during this period will not be taken into consideration in our assessment of the novelty presented by your study ("scooping" protection). We have extended this 'scooping protection policy' beyond the usual 3 month revision timeline to cover the period required for a full revision to address the essential experimental issues. Please contact me if you see a paper with related content published elsewhere to discuss the appropriate course of action.

Again, please contact me at any time during revision if you need any help or have further questions.

Thank you very much again for the opportunity to consider your work for publication. I look forward to your revision.

Best regards,

William

William Teale, Ph.D.
Editor
The EMBO Journal

When submitting your revised manuscript, please carefully review the instructions below and include the following items:

- 1) a .docx formatted version of the manuscript text (including legends for main figures, EV figures and tables). Please make sure that the changes are highlighted to be clearly visible.
- 2) individual production quality figure files as .eps, .tif, .jpg (one file per figure).
- 3) a .docx formatted letter INCLUDING the reviewers' reports and your detailed point-by-point response to their comments. As part of the EMBO Press transparent editorial process, the point-by-point response is part of the Review Process File (RPF), which will be published alongside your paper.
- 4) a complete author checklist, which you can download from our author guidelines ([https://wol-prod-cdn.literatumonline.com/pb-assets/embo-site/Author Checklist%20-%20EMBO%20J-1561436015657.xlsx](https://wol-prod-cdn.literatumonline.com/pb-assets/embo-site/Author%20Checklist%20-%20EMBO%20J-1561436015657.xlsx)). Please insert information in the checklist that is also reflected in the manuscript. The completed author checklist will also be part of the RPF.
- 5) Please note that all corresponding authors are required to supply an ORCID ID for their name upon submission of a revised manuscript.
- 6) We require a 'Data Availability' section after the Materials and Methods. Before submitting your revision, primary datasets produced in this study need to be deposited in an appropriate public database, and the accession numbers and database listed

under 'Data Availability'. Please remember to provide a reviewer password if the datasets are not yet public (see <https://www.embopress.org/page/journal/14602075/authorguide#datadeposition>). If no data deposition in external databases is needed for this paper, please then state in this section: This study includes no data deposited in external repositories. Note that the Data Availability Section is restricted to new primary data that are part of this study.

Note - All links should resolve to a page where the data can be accessed.

8) For data quantification: please specify the name of the statistical test used to generate error bars and P values, the number (n) of independent experiments (specify technical or biological replicates) underlying each data point and the test used to calculate p-values in each figure legend. The figure legends should contain a basic description of n, P and the test applied. Graphs must include a description of the bars and the error bars (s.d., s.e.m.).

9) We would also encourage you to include the source data for figure panels that show essential data. Numerical data can be provided as individual .xls or .csv files (including a tab describing the data). For 'blots' or microscopy, uncropped images should be submitted (using a zip archive or a single pdf per main figure if multiple images need to be supplied for one panel). Additional information on source data and instruction on how to label the files are available at .

10) We replaced Supplementary Information with Expanded View (EV) Figures and Tables that are collapsible/expandable online (see examples in <https://www.embopress.org/doi/10.15252/embj.201695874>). A maximum of 5 EV Figures can be typeset. EV Figures should be cited as 'Figure EV1, Figure EV2" etc. in the text and their respective legends should be included in the main text after the legends of regular figures.

12) Our journal encourages inclusion of *data citations in the reference list* to directly cite datasets that were re-used and obtained from public databases. Data citations in the article text are distinct from normal bibliographical citations and should directly link to the database records from which the data can be accessed. In the main text, data citations are formatted as follows: "Data ref: Smith et al, 2001" or "Data ref: NCBI Sequence Read Archive PRJNA342805, 2017". In the Reference list, data citations must be labeled with "[DATASET]". A data reference must provide the database name, accession number/identifiers and a resolvable link to the landing page from which the data can be accessed at the end of the reference. Further instructions are available at .

13) In order to increase the reproducibility and reach of your work, The EMBO Journal includes a table of reagents that were used in the study. Please provide this along with your revisions.

Further instructions for preparing your revised manuscript:

- a point-by-point response to the referees' comments, with a detailed description of the changes made (as a word file).
- a word file of the manuscript text.
- individual production quality figure files (one file per figure)
- a complete author checklist, which you can download from our author guidelines (<https://www.embopress.org/page/journal/14602075/authorguide>).

- Expanded View files (replacing Supplementary Information)

We realize that it is difficult to revise to a specific deadline. In the interest of protecting the conceptual advance provided by the work, we recommend a revision within 3 months (12th Mar 2025). Please discuss the revision progress ahead of this time with the editor if you require more time to complete the revisions. Use the link below to submit your revision:

Referee #1:

In this manuscript, Gao et al. describe a novel role for the protein Copine-6 in sensory neurons. In particular, they provide molecular and cellular evidence that Copine-6 promotes translocation of the cation channel TRPM3, a known heat and pain sensor, to the plasma membrane. In addition, they provide ion vivo behavioral measurements in mice and rats, which show that knockdown or knockout of Copine-6 leads to a specific increase in the response latency to heat stimuli, as well as lower pain responses to injection of TRPM3 agonist.

Overall, this work is interesting and potentially of significant importance, as it provides (1) novel insights into the biological roles of Copine-6, and (2) a new regulatory mechanism for TRPM3, a promising new pain target. There are, however, some aspects to the study that deserve some attention.

1) The authors show that knockdown/knockout of Copine-6 has effects on normal heat sensitivity and heat hypersensitivity following CFA treatment, and attribute this to suppression of TRPM3 activity. Some caution is needed here, since the effects of Copine-6 knockdown on acute heat sensing appear to be at least as robust as full TRPM3 knockout, whereas, on the functional level, the reduction in TRPM3 currents/calcium signals in the copine-6 knockout neurons is relatively limited (~50% or less - figure 6). This begs the question whether Copine-6 may have additional effects in the neurons that contribute to the reduced heat responses. One possible way to address this would be to test whether Copine-6 downregulation/overexpression has an effect on behavioral heat responses under conditions where TRPM3 is suppressed (using antagonists) or in TRPM3 KO animals.

2) One possible additional explanation for the effect of Copine-6 could be an effect on overall neuronal excitability. In this respect, I noticed that the final calcium level upon potassium stimulation (Figure 6D) is lower in the Copine-6 knockouts. Additional data on neuronal excitability in the absence/presence of Copine-6 need to be included to evaluate this.

3) It is surprising that the authors did not test the effect of Copine-6 on heat responses of TRPM3. Given the complex gating behavior of this channel, one possibility could be that the effect on heat sensitivity (or endogenous ligands such as PregS) is more pronounced than on the responses to CIM0216. I would therefore advise the authors to test whether Copine-6 expression/knockdown affects heat responses in HEK293 cells/DRG neurons, and overall whether the number and amplitude of

heat-responsive neurons in DRG cultures is affected by Copine-6.

3) There are several splice variants of TRPM3, some of which show different pharmacological properties or altered modulation by cytosolic partners such as Gbetagamma. It is unclear whether the described mechanism is valid for all isoforms. Overall, it would be important to better delineate (e.g. binding site on the channel) the interaction with the channel, and to better describe the functional consequences (e.g. calcium dependence, heat sensitivity, voltage dependence).

Referee #2:

Gao and colleagues study the role of the copine-6 in DRG neurons, where this protein is highly expressed. They find that shRNA mediated downregulation of copine6 in DRG neurons in rats reduces heat sensitivity, but does not affect other sensory modalities. Copine6 KO mice have similar phenotype, which is reversed by DRG specific re-expression of copine 6. They identify TRPM3, a heat sensitive ion channel, as the target of copine 6, and show that copine6 increases trafficking of TRPM3 to the plasma membrane. Overall the work is important and appropriate for publication in EMBO Journal, because it identifies a novel regulator of TRPM3 and heat sensitivity, but there are a number of concerns, which need to be addressed.

Major concerns

1. Figure 2B on the level of downregulation of copine6 by shRNA injection needs to be quantified. The image shows almost no expression of copine6, which is in discrepancy with the Western blot and Fig EV1G. In Figure 1B, there is almost no immunofluorescence after shRNA injection, whereas in the summary of the Western blot in Fig EV1G show ~50 percent reduction. I think the level of downregulation is important for the interpretation of the data and it should be clear from a main figure, without the reader having to open the EV figures.

2. In page 8 the authors state: "The analysis revealed a high degree of overlap of expression for Cpne6 with Trpm3 in the same subtypes of DRG neurons, both in mouse and human DRG". This is not what the data show in (Fig. 4AB). There is some co-expression but it is nowhere near high degree. For example, in mouse neurons, PEP1 and NP1-3 shows high level of TRPM3, and almost no copine 6. In the view of this, I am quite concerned about the almost perfect colocalization of Copine6 and TRPM3 in panel E. It brings in the question of specificity of the TRPM3 antibody. Antibodies against ion channels are notoriously bad, and unless the authors validate the specificity of their antibody in their hands in DRG neurons from TRPM3 knockout mice, I remain skeptical of those data. The co-expression of TRPM3 with copine6 should be quantified with an alternative method, such as RNAscope. If the authors have TRPM3 knockout mice, they should validate their antibody, as it was used for other key experiments also, for example next point

3. The PLA experiment is performed without any negative controls, so we do not know if the dots are background, or reflect actual interaction. For example, an unrelated protein that does not interact with TRPM3 would be a good negative control. Also, the figure showing the PLA data should be cited.

4. Figure 7C shows increased membrane TRPM3 protein when co-expressed with copine. This is confusing, as membrane may also include ER, Golgi, etc. Please clarify throughout if you mean plasma membrane. According to the methods, they use a plasma membrane fraction isolation kit. Separating ER and other membranes from plasma membrane is not trivial, and I am not sure how reliable this kit is and how much is the plasma membrane is enriched by this kit. I would be more confident in the data if the authors verified these findings with the widely used cell-surface biotinylation assay.

Minor concerns

1. Is this the first copine6 KO mouse ? If yes, please provide a general description, i.e. are they born in mendelian ratios, and have normal fertility, are they generally healthy, do they have any obvious behavioral or morphological defects, etc.

2. TRPM3 is also expressed in the brain, and in humans its gain of function phenotype is associated with neurodevelopmental disorders which includes cognitive deficits and epilepsy (PMID 32343227 & 36648066). Even though not directly related to the topic of the paper, this should be briefly discussed to orient the reader not familiar with the topic. Especially that copine6 is also highly expressed in the brain.

3. Was siRNA transfection efficiency verified, for example was the siRNA fluorescently labeled ? DRG neurons are not easy to transfect, and the siRNA transfection showed 50% decrease in RNA level, which is only possible if transfection efficiency was at least 50%.

4. In Figure EV2A there seem to be no hypersensitivity after SNI. Please discuss and clarify.

5. The authors reference the Allen Spinal Cord atlas. Do they mean that copine6 is expressed in the spinal cord? TRPM3 is also expressed in spinal cord neurons (e.g. PMID: 33478988). Maybe the authors can briefly discuss.

6. The authors may consider an agonist dose-response with and without copine 6, to see if there is also an effect on agonist sensitivity, or the effect is entirely due to increase surface expression. Pregnenolone Sulfate may be more suitable for this purpose as it acts faster than CIM0216.
7. It is unclear what is plotted in Figure 6D. Is this all neurons from a coverslip, a mouse, or average of all data? It is important to clarify, as effect of copine6 in panel D is a lot larger than in the summary in panel E.
8. For the HEK cell experiments, did the authors test if there is endogenous copine 6 expression in these cells ?
9. Page 10: the number of nocifensive reaction to capsaicin is lower than that to CIM0216. This is the opposite of what most other labs find, and the responses to capsaicin are much smaller than generally seen on the literature. Please comment.
10. Given the potential concerns with TRPM3 antibodies, maybe the authors can verify the increased surface localization using their tomato-tagged TRPM3 construct in confocal microscopy or TIRF experiments.

Additional non-essential suggestions

1. Did any of the other copine isoforms that are expressed in DRG neurons have any effect on TRPM3? What is the level of sequence identity to copine 6, especially in the vWA domain?

We are thankful to the Reviewers and the Editor for thorough and positive evaluation of our study, and for their helpful suggestions. We believe that addressing these insightful comments has made the manuscript more compelling. Below is our full response to all comments raised.

Referee #1:

1) The authors show that knockdown/knockout of Copine-6 has effects on normal heat sensitivity and heat hypersensitivity following CFA treatment, and attribute this to suppression of TRPM3 activity. Some caution is needed here, since the effects of Copine-6 knockdown on acute heat sensing appear to be at least as robust as full TRPM3 knockout, whereas, on the functional level, the reduction in TRPM3 currents/calcium signals in the copine-6 knockout neurons is relatively limited (~50% or less - figure 6). This begs the question whether Copine-6 may have additional effects in the neurons that contribute to the reduced heat responses. One possible way to address this would be to test whether Copine-6 downregulation/overexpression has an effect on behavioral heat responses under conditions where TRPM3 is suppressed (using antagonists) or in TRPM3 KO animals.

The reviewer is making a valid point, and we will address it below with a commentary and new experiments. Firstly, we did not use TRPM3 knockout mice in this study, so we believe the reviewer is concerned about similar effect on heat sensitivity in rats with Copine-6 knocked down by viral shRNA and in *Cpne6*^{-/-} KO mice. We would like to point out two things: i) KD and KO is in different species; ii) while shRNA KD is partial, the full knockout of *Cpne6* in mice may have resulted in compensation from other genes, as is often the case with full knockouts. Hence, it is likely that both the KD and KO resulted in incomplete phenotype, hence is the similarity in the magnitude of effect on heat sensitivity.

To address this concern experimentally, we performed new experiments where noxious heat sensitivity was tested in the WT and *Cpne6*^{-/-} mice with and without i.p. injection of TRPM3 antagonist, isosakuranetin (2 mg/kg). The antagonist reduced heat sensitivity in the WT but not in the KO animals. Of note, the degree of increase in noxious heat withdrawal latency (Hargreaves test) induced by TRPM3 antagonist and by *Cpne6*^{-/-} KO was very similar. See below and Fig. 6K in the revised manuscript.

We believe that while Copine-6 may have other effects and interactions elsewhere, the modulation of heat sensitivity is mainly due to its interactions with TRPM3.

2) One possible additional explanation for the effect of Copine-6 could be an effect on overall neuronal excitability. In this respect, I noticed that the final calcium level upon potassium stimulation (Figure 6D) is lower in the Copine-6 knockouts. Additional data on neuronal excitability in the absence/presence of Copine-6 need to be included to evaluate this.

We have now performed general excitability tests on DRG neurons from WT and *Cpne6*^{-/-} KO mice (below and Fig. EV9) and did not find major effects of Copine-6 KD. Besides, Since Copine-6 expression is not limited to TRPM3-expressing neurons, a ‘general increase in excitability’ would likely to manifest in sensitization of other sensory modalities (noxious and innocuous touch, cold, etc.), which is clearly not the case here. Also, regarding the Fig. 6D data, these are normalised levels (F/F₀) and the relative increase in Ca²⁺ induced by high-K⁺ solution (from the previous plateau) is quite similar between two genotypes, indicating similar excitability.

3) It is surprising that the authors did not test the effect of Copine-6 on heat responses of TRPM3. Given the complex gating behavior of this channel, one possibility could be that the effect on heat sensitivity (or endogenous ligands such as PregS) is more pronounced than on the responses to CIM0216. I would therefore advise the authors to test whether Copine-6 expression/knockdown affects heat responses in HEK293 cells/DRG neurons, and overall whether the number and amplitude of heat-responsive neurons in DRG cultures is affected by Copine-6.

We agree, this was a shortcoming, owing to some difficulties with the precise and reliable temperature control on our patch-clamp rig. We have now overcome this technical challenge and

included characterization of the effect of *Cpne6* KO on heat responses of the DRG neurons (below and Fig. 6G-I).

4) There are several splice variants of TRPM3, some of which show different pharmacological properties or altered modulation by cytosolic partners such as Gbetagamma. It is unclear whether the described mechanism is valid for all isoforms. Overall, it would be important to better delineate (e.g. binding site on the channel) the interaction with the channel, and to better describe the functional consequences (e.g. calcium dependence, heat sensitivity, voltage dependence).

This is an important point. The electrophysiology data presented in the original manuscript were obtained with $\gamma 3$ splice variant of TRPM3, one of the four variants present in the DRG neurons, others are $\alpha 2$, $\alpha 3$ and $\gamma 2$ (Uchida et al. J Physiol Sci. 2019 69:623-634). We have now tested $\alpha 2$ and $\alpha 3$ and these are both similarly upregulated in the presence of Copine-6 (below and Fig. EV7).

We did not test $\gamma 2$, but it is almost identical to $\gamma 3$, with the only difference that it is missing exon 15, which is also absent in $\alpha 2$. Hence, we believe that the three variants we tested cover all the sequence variability of the DRG variants.

We also performed additional characterization of the effects of Copine-6 on TRPM3 voltage dependence and agonist binding. These data are presented in revised Fig. EV7 and below. Neither of these parameters were significantly affected by Copine-6, which is consistent with our hypothesis that Copine-6 regulates trafficking and membrane abundance of TRPM3 but not its ion channel activity.

We agree that even more functional tests could be included and that elucidation of the copine-6 binding site on TRPM3 is also an important line of enquiry, but given the scope of the present study we believe it would be more appropriate to pursue these in the future investigations.

Referee #2:

Major concerns

1. Figure 2B on the level of downregulation of copine6 by shRNA injection needs to be quantified. The image shows almost no expression of copine6, which is in discrepancy with the Western blot and Fig EV1G. In Figure 1B, there is almost no immunofluorescence after shRNA injection, whereas in the summary of the Western blot in Fig EV1G show ~50 percent reduction. I think the level of downregulation is important for the interpretation of the data and it should be clear from a main figure, without the reader having to open the EV figures.

We now quantified the IF data and including these in the revised Fig. 2. We think there could have been a pdf conversion or display issue, because the images for Fig. 2B clearly show lowered but very discernible Copine-6 immunoreactivity, see below.

Most importantly, close inspection of these images very clearly reveal, that in the case of scrambled shRNA (panel A), many neurons express both, GFP and Copine-6, while in the case of Copine-6 shRNA, there is an obvious mutual exclusion: ‘green’ cells expressing shRNA do not display Copine-6 immunoreactivity while many cells that do not express GFP (and shRNA) do still express Copine-6. Hence, we believe these images speak for themselves: the knockdown works and in the intended way.

2. In page 8 the authors state: "The analysis revealed a high degree of overlap of expression for Cpne6 with Trpm3 in the same subtypes of DRG neurons, both in mouse and human DRG". This is not what the data show in (Fig. 4AB). There is some co-expression but it is nowhere near high degree. For example, in mouse neurons, PEP1 and NP1-3 shows high level of TRPM3, and almost no copine 6.

We apologise for the poor wording, what we meant is that amongst the four heat-sensitive channels, *Trpm3* shows the highest degree of overlap with *Cpne6*. We have reworded this statement in the revised manuscript. Please see below with red squares highlighting subpopulations with highest *Cpne-6* expression:

Figure 4A-B

In the view of this, I am quite concerned about the almost perfect colocalization of Copine6 and TRPM3 in panel E. It brings in the question of specificity of the TRPM3 antibody. Antibodies against ion channels are notoriously bad, and unless the authors validate the specificity of their antibody in their hands in DRG neurons from TRPM3 knockout mice, I remain skeptical of those data. The co-expression of TRPM3 with copine6 should be quantified with an alternative method, such as RNAscope. If the authors have TRPM3 knockout mice, they should validate their antibody, as it was used for other key experiments also, for example next point

We would like to defend the images shown in Fig. 4E. We respectfully disagree with the assessment that there is ‘almost perfect’ colocalization between, TRPM3 and Copine-6. In fact, on the overlay image reproduced below one can clearly see that while there is a good number of cells expressing both proteins (appear yellow), many small neurons that are TRPM3-positive (red) have no/low Copine-6 immunofluorescence, while there are plenty of Copine-6-positive cells (green) that have no/low TRPM3 immunofluorescence. Thus, these stainings are in good agreement with the transcriptomic analysis shown in Fig. 4A, B. We realised that Venn diagram quantifying the colocalization (Fig. 4E) was inadvertently distorted during figure preparation (the numbers of cells in it are correct though); this may have contributed to the impression that there is an almost complete overlap between TRPM3 and Copine-6 immunofluorescence. This is now corrected, and all Venn diagrams are carefully checked in the revised figures. We apologise for this mistake.

We do not have TRPM8 KO, but we performed validation of TRPM3 and Copine-6 antibodies using CHO cells overexpressing RFP-tagged TRPM3 and flag-tagged Copine-6. These stainings revealed that our antibody only label cells expressing the corresponding epitope; the data are included in the new Fig. EV5.

Finally, we are now confirmed the co-expression of Copine-6 and TRPM3 using RNAscope, as requested. These data are now included in Fig. EV6. RNAscope data returned very similar pattern of co-expression, cf. Fig. 4E and Fig. EV6.

3. The PLA experiment is performed without any negative controls, so we do not know if the dots are background, or reflect actual interaction. For example, an unrelated protein that does not interact with TRPM3 would be a good negative control. Also, the figure showing the PLA data should be cited.

This is a good suggestion, we performed PLA between Copine-6 and TRPV1, a channel which display no functional modulation by Copine-6 in our studies (Fig. 5C, Fig. 6G, Fig. 8C). The PLA between Copine-6 and TRPV1 was mostly negative and these data are now included in revised Fig. 9.

4. Figure 7C shows increased membrane TRPM3 protein when co-expressed with copine. This is confusing, as membrane may also include ER, Golgi, etc. Please clarify throughout if you mean plasma membrane. According to the methods, they use a plasma membrane fraction isolation kit. Separating ER and other membranes from plasma membrane is not trivial, and I am not sure how reliable this kit is and how much is the plasma membrane is enriched by this kit. I would be more confident in the data if the authors verified these findings with the widely used cell-surface biotinylation assay.

We used a popular and well-cited kit, which was verified to successfully separate plasma membrane from other cellular membranes (see below). Importantly, we confirm these data with an independent method – confocal microscopy (Fig. 8).

Minute™ Plasma Membrane Protein Isolation and Cell Fractionation Kit

430 citations

参考文献: SM-005文献列表

424.Zhaoying Zhang; Caiyue Rier; Rong Xiao; Shuaiya Ma; H Liu; et al.(2024).Palmitoylation of TIM-3 promotes immune exhaustion and restrains antitumor immunity.Science Immunology.DOI 10.1126/sciimmunol.adg7302

425.Brajendra K. Tripathi; Nicole Hirsch; Xiaodan Qian; Marian E. Durkin; Dunnu Wang; et al.(2024).The pro-oncogenic noncanonical activity of a RAS-GTP-RanGAP1 complex facilitates nuclear protein export.Nature cancer.DOI 10.1038/s43018-024-00847-5

426.Martina Massarotti; Paola Coma; Archana Malik; Gloria Milanese; Claudio Casali; et al.(2024).Development and Biological Characterization of Cancer Biomimetic Membrane Nanovesicles for Enhancing Therapy Efficacy in Human Glioblastoma Cells.Nanomaterials.DOI 10.3390/nano14221779

427.Anusree Sasidharan; Astrid Grosche; Xiaodong Xu; T. Bernard Kinane; Damiano Angoli; et al.(2024). Select amino acids recover cytokine-altered ENaC function in human bronchial epithelial cells. PLoS ONE.DOI 10.1371/journal.pone.0307809

428.Meng-Chieh Lin; Wen-Hung Kuo; Shih-Yin Chen; Jing-Ya Hsu; Li-Yu Lu; et al.(2024).Ago2/CAV1 interaction potentiates metastasis via controlling Ago2 localization and miRNA action.EMBO Reports.DOI 10.1038/s44319-024-00132-7

429.Zengqi Tan; Lulu Ning; Lin Cao; Yue Zhou; Jing Li; et al.(2024).Bisecting GlcNAc modification reverses the chemoresistance via attenuating the function of P-gp.Theranostics.DOI 10.7150/thno.93879

430.Liu P., Sun L., Zhang Y., Tan Y., Zhu Y., Peng C., Wang J., Yan H., Mao D.,Liang G., Liang G., Li X., Liang Y., Wang F., He Z., Tang W., Huang D., and Chen C. (2024). The metal tolerance protein OsMTP11 facilitates cadmium sequestration in the vacuoles of leaf vascular cells for restricting its translocation into rice grains. Mol. Plant. doi: <https://doi.org/10.1016/j.molp.2024.09.012>.

Minor concerns

1. Is this the first copine6 KO mouse? If yes, please provide a general description, i.e. are they born in mendelian ratios, and have normal fertility, are they generally healthy, do they have any obvious behavioral or morphological defects, etc.

This is the first copine-6 KO mouse. The mice were born at Mendelian rates, exhibited normal fertility and were generally healthy. They also demonstrated normal motor capacity and coordination and showed no significant behavioural or morphological defects. This is now added to the revised Methods.

2. TRPM3 is also expressed in the brain, and in humans its gain of function phenotype is associated with neurodevelopmental disorders which includes cognitive deficits and epilepsy (PMID 32343227 & 36648066). Even though not directly related to the topic of the paper, this should be briefly discussed to orient the reader not familiar with the topic. Especially that copine6 is also highly expressed in the brain.

This is a good suggestion and we have added a comment about this to the discussion.

3. Was siRNA transfection efficiency verified, for example was the siRNA fluorescently labeled? DRG neurons are not easy to transfect, and the siRNA transfection showed 50% decrease in RNA level, which is only possible if transfection efficiency was at least 50%.

We only used siRNA in one set of experiments, to verify the effectiveness of this target sequence in cultured DRG neurons (Fig. EV1D). Everywhere else in the manuscript we used virally-delivered shRNA, with the same target sequence. siRNA is not difficult to transfect in DRG neurons (in contrast to cDNA).

4. In Figure EV2A there seem to be no hypersensitivity after SNI. Please discuss and clarify.

There was a trend towards hypersensitivity in both genotypes but it did not reach significance. A comment has been added to the Results

5. The authors reference the Allen Spinal Cord atlas. Do they mean that copine6 is expressed in the spinal cord? TRPM3 is also expressed in spinal cord neurons (e.g. PMID: 33478988). Maybe the authors can briefly discuss.

Allen Spinal cord atlas includes DRGs in samples from younger animals, please see below.

6. The authors may consider an agonist dose-response with and without copine 6, to see if there is also an effect on agonist sensitivity, or the effect is entirely due to increase surface expression. Pregnenolone Sulfate may be more suitable for this purpose as it acts faster than CIM0216.

We have performed these experiments, as suggested and these are now presented in the revised Fig. EV7 (see also response to point 4 of referee #1). There was no significant change in agonist sensitivity (we tested both, PregS and CIM0216).

7. It is unclear what is plotted in Figure 6D. Is this all neurons from a coverslip, a mouse, or average of all data? It is important to clarify, as effect of copine6 in panel D is a lot larger than in the summary in panel E.

Shown in this panel are averages of all neurons from one field of view. This is now clarified in the legend.

8. For the HEK cell experiments, did the authors test if there is endogenous copine 6 expression in these cells ?

We tested and it is undetectable; see below and Fig. EV1B

9. Page 10: the number of nocifensive reaction to capsaicin is lower than that to CIM0216. This is the opposite of what most other labs find, and the responses to capsaicin are much smaller than generally seen on the literature. Please comment.

This was indeed surprising, we re-ran these experiments with fresh capsaicin and obtained higher nocifensive responses, hence, we think that reagent we originally used may have degraded. We replaced these data now and re-analysed as time spent licking and lifting the paw, rather than number of these events, to be more in line with the literature. See below and Fig. 6J.

10. Given the potential concerns with TRPM3 antibodies, maybe the authors can verify the increased surface localization using their tomato-tagged TRPM3 construct in confocal microscopy or TIRF experiments.

We verified the antibodies for TRPM3 and Copine-6. We overexpressed TRPM3-RFP or Copine-6-flag in CHO cells and showed that antibody only binds to red or flag-positive cells.

Additional non-essential suggestions

1. Did any of the other copine isoforms that are expressed in DRG neurons have any effect on TRPM3? What is the level of sequence identity to copine 6, especially in the vWA domain?

The most expressed in rat DRG are Copine-6, 3 and 4 (see below). We have data that Copine-3 does not influence heat sensitivity, these are a part of a separate study and we prefer not to disclose these data here. We did not test Copine-4 yet, the overall sequence identity of this isoform with Copine-6 is 61% and 67% within the vWA domain, so while both copines are indeed similar, they are not identical and further studies will be required to elucidate specific functions of individual isoforms.

Editor's suggestion: the Editor suggested to test if *Cpne6* shRNA affects expression of other relevant *Cpne6* isoforms in the DRG. We have performed these experiments and these are now included in the revised Fig. EV1E, F (and below). There was no effect on the level of expression of other copines in the rat DRG.

Additional changes: additional data were added to Fig.8 analysis to increase robustness of this dataset. This did not change the conclusions of these analyses.

Dear Nikita,

We have now received re-review reports from both referees, which I have included below. As you will see, you have addressed their concerns satisfactorily. However, before I can finally accept the manuscript, there are some remaining editorial points which need to be addressed. In this regard would you please:

- acknowledge the following funders on our online submission system: the Hebei Province Talent and Intelligence Introduction Project Award; Central Guiding Local Science and Technology Development Fund Project (236Z7723G); Hebei Natural Science Foundation award (H2022206515); Key laboratory of Neural and Vascular Biology, Ministry of Education of China project NV20230001,
- rename the conflict of interest section as the 'Disclosure and Competing Interests' section,
- remove the author credit section from the manuscript,
- we no longer use references to 'data not shown'. On page 8, I suggest 'As TRPM2 immunofluorescence was also weak and widely distributed in the DRG, it was excluded from co-localization analysis with Copine-6',
- ensure all figure callouts are listed sequentially; include a callout for Fig. 3E,
- upload the Reagents and Tools table as a separate file using the template from our guide to authors,
- supply measurements as source data for Fig. 6C and 6H; source data files need to be saved in a scheme of one figure/folder and then uploaded as .zip files. E.g. all the Source data files for figure 1 need to be saved in a single folder and this needs to be zipped and then uploaded as "SD figure 1.zip" file. For EV and/or appendix figures, ZIP together all source data. Completed SD checklist should be uploaded as Related Manuscript File,
- provide exact p values in the legends of figures 2B, E, I; 3G, H, J; 5B, 6B, C, E, F, H, I, J, K; 7C, G; 8E-G; 9B, E; EV1 A, C, D, F, H; EV2 B, EV3 A, EV4 B-D, I-L; EV7 C, H,
- indicate what */ **/ ***/ **** represent; if this represents p value(s), - - indicate the statistical test used,
- define 'n' in the legends of figures 6D, G, K; EV4H-L; EV7 F, G.
- define error bars in the legends of figures 5B, D, F, H; 6B, D, E, G, I, K; 7C, D, G; 8E-G; 9B, E; EV1 A, C, D, E, F, H; EV3 A, B; EV4 A-G; H-L, M, N, O; EV7 C, F, G, H; EV8 B, D; EV9 B-D,
- limit the number of EV figures to five, putting the others in the appendix, and
- correct the section order as follows: Title page - Abstract & Keywords - Introduction - Results - Discussion - Methods - Data Availability - Acknowledgements - Disclosure and Competing Interests Statement - References - Figure Legends - Table(s) - Expanded View Figure Legends.

We include a synopsis of the paper (see <http://emboj.emboPress.org/>). Please provide me with a general summary image, a two sentence statement and 3-5 bullet points that capture the key findings of the paper.

I am looking forward to receiving your revised manuscript.

EMBO Press is an editorially independent publishing platform for the development of EMBO scientific publications.

Best wishes,

William

William Teale, PhD
Editor
The EMBO Journal
w.teale@embojournal.org

See also figure legend guidelines: <https://www.emboPress.org/page/journal/14602075/authorguide#figureformat>

- a point-by-point response to the referees' comments, with a detailed description of the changes made (as a word file).
- a word file of the manuscript text.

- individual production quality figure files (one file per figure)
- a complete author checklist, which you can download from our author guidelines (<https://www.embopress.org/page/journal/14602075/authorguide>).
- Expanded View files (replacing Supplementary Information)

We realize that it is difficult to revise to a specific deadline. In the interest of protecting the conceptual advance provided by the work, we recommend a revision within 3 months (13th Aug 2025). Please discuss the revision progress ahead of this time with the editor if you require more time to complete the revisions. Use the link below to submit your revision:

Referee #2:

The authors provided satisfactory responses to the critiques, therefor I recommend acceptance

We are pleased that reviewers were happy with the revisions made to the previous version of the manuscript. Below we address editorial points raised by the Editor.

- acknowledge the following funders on our online submission system: the Hebei Province Talent and Intelligence Introduction Project Award; Central Guiding Local Science and Technology Development Fund Project (236Z7723G); Hebei Natural Science Foundation award (H2022206515); Key laboratory of Neural and Vascular Biology, Ministry of Education of China project NV20230001,

Done

- rename the conflict of interest section as the 'Disclosure and Competing Interests' section,

Done

- remove the author credit section from the manuscript

Done

- we no longer use references to 'data not shown'. On page 8, I suggest 'As TRPM2 immunofluorescence was also weak and widely distributed in the DRG, it was excluded from co-localization analysis with Copine-6',

Edited, as suggested

- ensure all figure callouts are listed sequentially; include a callout for Fig. 3E

The sequence of figure callouts has been checked and the necessary amendments made, this required some small rearrangements in text and figures, these are highlighted; Fig. 3E is now referenced.

- upload the Reagents and Tools table as a separate file using the template from our guide to authors

Done

- supply measurements as source data for Fig. 6C and 6H; source data files need to be saved in a scheme of one figure/folder and then uploaded as .zip files. E.g. all the Source data files for figure 1 need to be saved in a single folder and this needs to be zipped and then uploaded as "SD figure 1.zip" file. For EV and/or appendix figures, ZIP together all source data. Completed SD checklist should be uploaded as Related Manuscript File,

All this has been done

- provide exact p values in the legends of figures 2B, E, I; 3G, H, J; 5B, 6B, C, E, F, H, I, J, K; 7C, G; 8E-G; 9B, E; EV1 A, C, D, F, H; EV2 B, EV3 A, EV4 B-D, I-L; EV7 C, H,

- indicate what */ **/ ***/ **** represent; if this represents p value(s), - - indicate the statistical test used

Done, the exact p values are stated if larger than 0.001, for very small numbers (smaller than 0.001) p is stated as $p < 0.001$.

- define 'n' in the legends of figures 6D, G, K; EV4H-L; EV7 F, G.

Done

- define error bars in the legends of figures 5B, D, F, H; 6B, D, E, G, I, K; 7C, D, G; 8E-G; 9B, E; EV1 A, C, D, E, F, H; EV3 A, B; EV4 A-G; H-L, M, N, O; EV7 C, F, G, H; EV8 B, D; EV9 B-D

Done throughout.

- limit the number of EV figures to five, putting the others in the appendix

EV2, 3, 5, 6 are now removed to the Appendix (Fig S1-S4)

- correct the section order as follows: Title page - Abstract & Keywords - Introduction - Results - Discussion - Methods - Data Availability - Acknowledgements - Disclosure and Competing Interests Statement - References - Figure Legends - Table(s) - Expanded View Figure Legends.

Done

We include a synopsis of the paper (see <http://emboj.embopress.org/>). Please provide me with a general summary image, a two sentence statement and 3-5 bullet points that capture the key findings of the paper.

We have uploaded our synopsis as a separate file

Dear Nikita,

I am pleased to inform you that your manuscript has been accepted for publication in the EMBO Journal.

Congratulations to you and to all involved!

Best wishes,

William

William Teale, PhD
Editor
The EMBO Journal
w.teale@embojournal.org
